# Quantitative analysis of nuclear pore complex organization in *Schizosaccharomyces pombe*

Joseph M Varberg[1], Jay R Unruh[1], Andrew J Bestul[1], Azqa A Khan[1], Sue L Jaspersen[1,2]

**The number, distribution, and composition of nuclear pore complexes (NPCs) in the nuclear envelope varies between cell types and changes during cellular differentiation and in disease. To understand how NPC density and organization are controlled, we analyzed the NPC number and distribution in the fission yeast *Schizosaccharomyces pombe* using structured illumination microscopy. The small size of yeast nuclei, genetic features of fungi, and our robust image analysis pipeline allowed us to study NPCs in intact nuclei under multiple conditions. Our data revealed that NPC density is maintained across a wide range of nuclear sizes. Regions of reduced NPC density are observed over the nucleolus and surrounding the spindle pole body (SPB). Lem2-mediated tethering of the centromeres to the SPB is required to maintain NPC exclusion near SPBs. These findings provide a quantitative understanding of NPC number and distribution in *S. pombe* and show that interactions between the centromere and the nuclear envelope influences local NPC distribution.**

## Introduction

Nuclear pore complexes (NPCs) facilitate nucleocytoplasmic transport, organize the genome, influence gene expression, and facilitate DNA repair (Raices and D'Angelo, 2017; Lin & Hoelz, 2019; Pascual-Garcia & Capelson, 2021). Each NPC is composed of multiple copies of ~30 individual nucleoporin (Nup) proteins, which are organized around a central channel in eightfold symmetry (Stoffler et al, 2003; Beck, 2004; Alber et al, 2007; Beck & Hurt, 2017). The NPC is anchored in the nuclear envelope (NE) by transmembrane Nups and through interactions between specific Nups and lipids of the nuclear membrane. Decades of research in a variety of systems has identified conserved functions for Nups in NPC assembly and transport and has mapped their organization within the structure of the NPC at nearly atomic resolution (von Appen et al, 2015; Mosalaganti et al, 2018; Schuller et al, 2021; Zimmerli et al, 2021; Akey et al, 2022).

In contrast to our understanding of NPC structure, the mechanisms that control NPC density, distribution, and composition remain poorly understood. Early studies using EM showed that NPC density is highly variable between species and cell types (Maul, 1977; Maul & Deaven, 1977; Garcia-Segura et al, 1989). NPC density does not appear to correlate with nuclear size or DNA content, but it is associated with metabolic activity (Maul et al, 1980) perhaps explaining links between changes in NPC density and cancer (Czerniak et al, 1984; Sakuma et al, 2021) or in response to external signals (Maul et al, 1972; Many et al, 1981; Carmo-Fonseca, 1982; García-Segura et al, 1987; Ortiz & Cavicchia, 1990). The remarkably long half-lives of many Nups (D'Angelo et al, 2009; Savas et al, 2012; Toyama et al, 2013) has led to a proposal that the number of NPCs in a cell is likely regulated at the stage of NPC assembly (reviewed in Otsuka and Ellenberg [2018]). In metazoans and in budding yeast, the number of NPCs in the NE roughly doubles during interphase nuclear growth (Maul et al, 1972; Winey et al, 1997; Dultz & Ellenberg, 2010; Maeshima et al, 2011; Otsuka et al, 2016). NPC assembly during the cell cycle is positively regulated by cyclin-dependent kinases (Cdks) (Maeshima et al, 2010) and negatively regulated by phosphorylation of NPC assembly factors by extracellular signal–regulated kinase (ERK), which is recruited to the NPC by the basket Nup TPR (McCloskey et al, 2018). However, the ubiquitous role of TPR for negative regulation of NPC density has been debated (Boumendil et al, 2019; Kittisopikul et al, 2021).

Once assembled into the NE, NPCs adopt a variety of nonrandom distributions, ranging from pairs and clusters to higher order linear and hexagonal arrays (Maul, 1977). Plant, animal, and fungal nuclei have reduced NPC density in regions over the nucleolus and near sites of contact between the nucleus and cytosolic organelles, such as the vacuole/lysosome, Golgi apparatus, and mitochondrion (La Cour & Wells, 1974; Severs et al, 1976; Maul, 1977; Harris, 1978; Miller et al, 1995; Winey et al, 1997; Pan et al, 2000; Wang et al, 2016). Despite decades of work clearly demonstrating nonrandom NPC distributions in multiple cell types, little is known about how these patterns are formed and maintained. In metazoans, NPC distribution is mediated at least in part through the nuclear lamina (Aaronson & Blobel, 1974, 1975; Daigle et al, 2001; Kittisopikul et al, 2021). However, as both plants and fungi lack lamins, additional factors must serve to regulate NPC distribution. LAP2-emerin-MAN1 (LEM) domain proteins, which associate with the inner nuclear membrane (INM) throughout eukaryotes, are leading candidates and are enriched at

[1]Stowers Institute for Medical Research, Kansas City, MO, USA   [2]Department of Molecular and Integrative Physiology, University of Kansas Medical Center, Kansas City, KS, USA

Correspondence: jvarberg@stowers.org; slj@stowers.org

   

pore-free regions of the NE in cultured cells (Maeshima et al, 2006). In budding yeast, NPC density is increased in the region of the NE near the spindle pole body (SPB), suggesting that either the SPB itself or associated factors may recruit NPCs for NE remodeling during SPB insertion (Winey et al, 1997; Wang et al, 2016; Rüthnick et al, 2017). In contrast to *Saccharomyces cerevisiae*, in which the SPB remains embedded in the NE throughout the cell cycle, *Schizosaccharomyces pombe* SPBs remain cytoplasmic throughout interphase and are only inserted into the NE during mitosis (Jaspersen, 2021). Whether NPCs provide a similar function in SPB insertion in *S. pombe* remains unexplored.

Analysis of NPC composition in the region over the nucleolus in *S. cerevisiae* showed that nucleolar-associated NPCs lack two Nups, Mlp1 and Mlp2 (Strambio-de-Castillia et al, 1999; Galy et al, 2004) These orthologs of vertebrate Nup Tpr (translocated promoter region) are core structural components of the nuclear basket, a nucleoplasmic extension of the NPC that serves as a binding site for chromatin, proteasomes, and other factors (Bangs et al, 1998; Bae et al, 2009; Niepel et al, 2013; Salas-Pino et al, 2017). These data clearly demonstrate that *S. cerevisiae* maintains multiple, compositionally distinct populations of NPCs in specific subregions of the NE. How these populations are established remains unclear, although recent work has implicated mRNA transcription and processing in Mlp1 recruitment (Bensidoun et al, 2021 *Preprint*). Furthermore, it is unknown whether this is a unique property of budding yeast nuclei, as may be the case for other aspects of its NPC biology including mechanisms controlling inheritance of NPCs during mitotic divisions (which rely on the *S. cerevisiae* bud neck structure) (Shcheprova et al, 2008; Boettcher et al, 2012; Colombi et al, 2013; Makio et al, 2013; Kumar et al, 2018) and NPC remodeling during budding yeast meiosis (which is not seen in *S. pombe*) (Asakawa et al, 2010; King et al, 2019). Whether these patterns of NPC heterogeneity are conserved in *S. pombe* is of particular interest as the number of nucleoplasmic Y-complex rings and organization of the cytoplasmic rings differs between budding and fission yeast (Zimmerli et al, 2021). In addition, identifying the mechanisms that control heterogeneity in NPC composition and distribution are of great interest as transcriptomic and proteomic studies in metazoans have identified cell type–specific Nup expression patterns and have shown that changes in NPC composition are critically important for cell development, differentiation, and progression of various diseases (D'Angelo et al, 2012; Ori et al, 2013; Gomez-Cavazos & Hetzer, 2015; Capitanchik et al, 2018; Kane et al, 2018; Guglielmi et al, 2020). These findings, in combination with the evidence for NPC compositional heterogeneity within individual nuclei in budding yeast, highlight the emerging concept that subpopulations of NPCs with distinct compositions and potentially specialized functions may exist at specific locations within the NE (Fernandez-Martinez & Rout, 2021; Akey et al, 2022).

Using *S. pombe* as a model system, we combined multiple quantitative imaging approaches, including three-dimensional structured illumination microscopy (3D-SIM), to examine the number, distribution, and composition of NPCs in whole nuclei. We quantify the NPC number under a range of conditions and show that fission yeast maintains a constant NPC density throughout its life cycle. NPC density appears to be maintained through a mechanism that links NPC assembly to increases in the available NE

surface area. Experiments using 3D-SIM and live-cell imaging revealed a common structural organization of NPC clusters and identified two distinct behaviors of clusters during mitotic cell division. We show that the previously reported reduction of NPC density and alteration of NPC basket composition over the nucleolus-facing region of the NE is conserved in fission yeast. In addition, NPCs are excluded from the NE region surrounding the SPBs by Lem2 and other factors.

# Results

### 3D-SIM image analysis pipeline for NPC quantitation

We developed an imaging and analysis pipeline to visualize and count NPCs in fission yeast after three-dimensional structured illumination microscopy (3D-SIM) of entire nuclei containing endogenously tagged Nups. This approach provides a roughly twofold increase in resolution as compared with conventional light microscopy, with lateral resolution that approaches the size of the yeast nuclear pore (~90–120 nm diameter) (Zimmerli et al, 2021; Akey et al, 2022). Individual foci corresponding to single NPCs (full width half maximum [FWHM] = 121.6–136.5 nm, 95% CI) and larger foci that likely represent clusters of NPCs that cannot be fully resolved by SIM were detected throughout the nucleus (Fig 1A). NPCs could be visualized using various tagged Nups, including representatives from each NPC subcomplex (Fig 1B). We observed strong correlation in the number and location of NPCs detected in both channels in strains co-expressing Nup40-mCherry and Nup44-GFP, confirming that our approach detects bona fide NPCs (Fig S1A–C). Measurements of NPC number and position, nuclear size, and cell cycle stage could be extracted from images using the strategy outlined in Fig 1C for the stages illustrated in Fig 1D and used to derive values for nuclear surface area and NPC density through the cell cycle (see the Materials and Methods section).

In *S. pombe*, nuclear size increases through the interphase to maintain a constant nuclear-to-cell volume ratio (Neumann & Nurse, 2007). Using multiple tagged Nups, we found that the number of NPCs also increases to maintain an NPC density that varies less than 10% through the cell cycle (Table S1 and Figs 1E and S1E). We observed that mother and daughter nuclei often had differences in NPC densities (8/20 pairs with ≥ 20% difference), reminiscent of the elevated NPC density observed in daughter nuclei for *S. cerevisiae* (Colombi et al, 2013). Visualization of Cdc7-GFP, a kinase that preferentially localizes to the "new" SPB during anaphase B (Grallert et al, 2004), showed that the asymmetric NPC densities we observed is random with respect to the inheritance of the "old" or "new" SPB (Fig S1D). During the late stages of *S. pombe* mitosis, a subset of NPCs localize to the membrane bridge where they facilitate active transport before being selectively disassembled in a temporal order, starting with removal of the basket, to trigger localized NE breakdown and spindle disassembly (Lucena et al, 2015; Dey et al, 2020; Expósito-Serrano et al, 2020). In agreement with these findings, we observed NPCs in the anaphase bridge midzone that contained transmembrane (Cut11) and structural nucleoporins (Nup37) but lacked the basket (Nup60) (Fig S1F). Because of their dynamic nature, bridge NPCs were excluded from our quantitative cell cycle measurements.

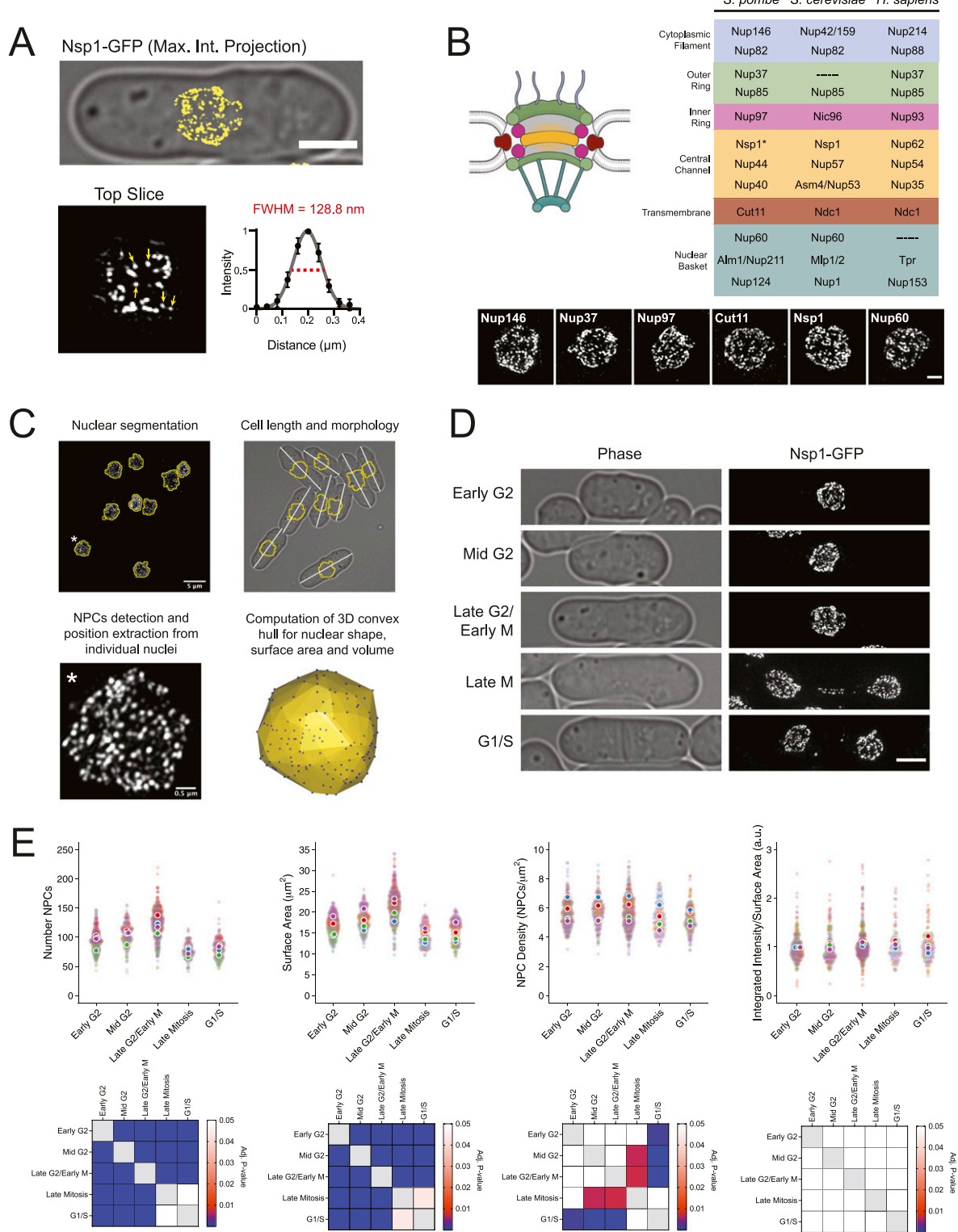

**Figure 1. A 3D-SIM imaging and analysis pipeline to measure the nuclear pore complex (NPC) number and density in *S. pombe*.**
**(A)** 3D-SIM image of Nsp1-GFP overlayed on transmitted light image. Bar, 3 μm. Arrows show five single NPCs fit to a Gaussian to generate the average NPC intensity profile (full-width half-maximum, red dashed line). **(B)** NPC with sub-complexes colored to match the table of Nups shown on right, with representative 3D-SIM images below. Bar, 1 μm. *Nsp1 present in channel and cytoplasmic complexes. **(C)** Pipeline for NPC analysis. **(D)** Representative 3D-SIM image from each cell cycle stage. Bar, 3 μm. **(E)** NPC number, nuclear surface area, NPC density, and area-normalized intensity measurements from four independent Nsp1-GFP replicates with replicate means indicated by color. Matrices showing all pair-wise comparisons shown below, based on Dunn's test.

Despite the improved lateral resolution offered by SIM, clustering of NPCs and the comparatively reduced axial resolution likely lead to undercounting of NPCs using 3D-SIM. To estimate the extent of undercounting, we applied our analysis pipeline to simulated datasets modeling randomly distributed NPCs for a range of NPC densities and nuclear sizes (Fig S1G). For the range of densities observed experimentally (~4–7 NPCs/$\mu m^2$), the measured values for NPC density and the NE surface area fell within 10–30% of the true simulated values for all simulated nuclear sizes, with the percent error increasing in a density-dependent manner. Because of the observed undercounting, we used a secondary approach that did not rely on segmentation of individual NPCs from 3D-SIM images to measure total Nup intensity as a proxy for NPC density. Nsp1-GFP intensities showed similar increases through the cell cycle, whereas Nup-GFP intensity per unit nuclear surface area remained constant (Fig 1E).

After correcting for the undercounting observed for our 3D-SIM approach, our Nsp1-GFP analysis estimates that the average mid-G2 stage fission yeast nucleus contains between 115 and 137 ± 22–26 NPCs, with a nuclear surface area of 18.4 ± 2.8 $\mu m^2$, for a density of 6.3–7.4 NPCs/$\mu m^2$ (n = 174). These values are lower than those reported for budding yeast (ranging from 9 to 15 NPCs/$\mu m^2$, [Moor & Mühlethaler, 1963; Maul 1977; Maul & Deaven, 1977; Winey et al, 1997]) but are similar to NPC densities reported for other cell types (ranging from 4.5 to 8 NPCs/$\mu m^2$) (Maul & Deaven, 1977; García-Segura et al, 1987; Garcia-Segura et al, 1989; Dultz & Ellenberg, 2010; Maeshima et al, 2010). Although similar trends in the NPC number and density were observed using multiple tagged nucleoporins (Fig S1E), variability in the number of NPCs detected was observed when comparing datasets between different tagged Nups and between replicates for the same tagged Nup (Figs 1E and S1H). Differences between Nups were reproducible but could not be attributed to differences in Nup intensity or to members of specific subcomplexes. As a result, all experiments comparing NPC densities between genetic backgrounds or treatment conditions were performed in cells expressing the same tagged Nup and have been normalized to emphasize the relative differences within each experiment.

### NPC density is controlled in a NE surface area–dependent manner

NPC density might be controlled by a mechanism coupling NPC assembly with the available NE surface area (McCloskey et al, 2018). To explore this possibility in *S. pombe*, we examined NPC density in cells with nuclei covering a broad range of sizes. Meiotic progeny, known as spores, have a similar NPC density to mitotic cells despite having nuclei with three- to fourfold lower nuclear surface area (Figs 2A and S2A). Similarly, a constant NPC density was maintained when nuclear size was reduced in mitotic cells using a temperature-sensitive allele of Wee1 kinase (*wee1.50*), a negative regulator of the cyclin-dependent kinase Cdk1/Cdc2 (Russell & Nurse, 1987) (Figs 2B and S2B). Cells expressing a temperature-sensitive mutation in the Cdk1/Cdc2–phosphatase Cdc25 (*cdc25.22*) are arrested at the G2/M boundary yet continue to increase both cell and nuclear size (Nurse et al, 1976). During *cdc25.22* arrest, both nuclear surface area and the number of NPCs roughly doubled over the 3.5 h incubation, allowing NPC density to be maintained (Figs 2C and S2C). The increase in the NPC number was dependent on NE membrane expansion during arrest as chemical inhibition of fatty-acid synthesis by treatment with cerulenin blocked nuclear growth, although NPC density was maintained (Fig 2A). Yeast lacking core components of the autophagy machinery (*atg8Δ* or *atg1Δ*) (Yorimitsu & Klionsky, 2005) that targets NPCs for degradation during nutrient deprivation do not show increased NPC density compared with wild-type cells, suggesting that autophagy is not likely to be used to as a negative regulator of NPC density in the absence of nutrient depletion (Figs 2D and S2D). These results support a model whereby NPC density is maintained by a mechanism that couples the assembly of new NPCs to increases in the NE surface area.

### NPC cluster organization and dynamics

Our ability to observe NPCs throughout entire nuclei using 3D-SIM at near single-NPC resolution allowed us to evaluate a higher level NPC organization. NPC clustering is a common phenotype in different cell types and in mutants defective in NPC assembly (Rout & Wente, 1994; Doye, 1995; Pappas et al, 2018; Cheng et al, 2021). Using 3D-SIM, we compared NPC distribution in wild-type cells to two previously described *S. pombe* clustering mutants: *nup132Δ* and *nem1Δ* (Baï et al, 2004; Asakawa et al, 2014; Makarova et al, 2016).

Widefield and confocal images of NPC clusters in *nup132Δ* mutants often appear as a few large clusters; however, 3D-SIM images revealed the presence of multiple smaller clusters distributed throughout the NE (Fig 3A). Most *nup132Δ* nuclei displayed normally distributed NPCs or very mild NPC clustering, with only 14% displaying moderate to severe clustering (Fig 3A). Unexpectedly, we frequently observed NPC clusters organized in a ring-like structure with diameters ranging from 200 to 300 nm (Fig 3B). In rare cases, ring-like NPC clusters were also observed in wild-type cells, suggesting that these are not simply a unique phenotype of *nup132+* deletion. Similar clusters were observed for multiple Nups in wild-type cells, suggesting that pores in these clusters contain structural (Nup37), central channel (Nsp1), basket (Nup60), and cytoplasmic filament (Nup146) components. Clustering increased in aged *nup132Δ* cells grown on plates (Fig 3C), consistent with previous reports (Baï et al, 2004). Similar rings were also observed in *nem1Δ* cells, which have increased rates of lipid synthesis that alters NE morphology and NPC distribution (Siniossoglou et al, 1998; Kim et al, 2007; Makarova et al, 2016; Dey et al, 2020) (Fig 3D).

To examine the dynamics of the NPC clusters through the cell cycle, we performed time-lapse imaging of *nup132Δ* and *nem1Δ* cells and monitored NPC clusters in single cells. These experiments revealed two surprisingly different behaviors for clustered NPCs. In *nem1Δ* mutants, NPC clustering became more severe as nuclei prepared to divide. NPC clusters were frequently enriched in the anaphase bridge, along with excess membrane (Figs 3E and S2E). After completion of nuclear division, the resulting daughter nuclei had normal NE morphologies and NPC densities equivalent to wild-type nuclei (Fig S2F). This suggests that *nem1Δ* nuclei can remove excess NE membranes and NPCs during mitosis via the anaphase bridge. In contrast, NPC clusters in *nup132Δ* nuclei coalesced into larger clusters that preferentially localized to the SPBs in mitosis (Fig 3F and G). SPB-associated clusters are then segregated into the mother and daughter nuclei as cells complete mitosis. Upon entry into G1, clusters are evident but are no longer enriched near the SPBs (Video 1). These observations suggest at least

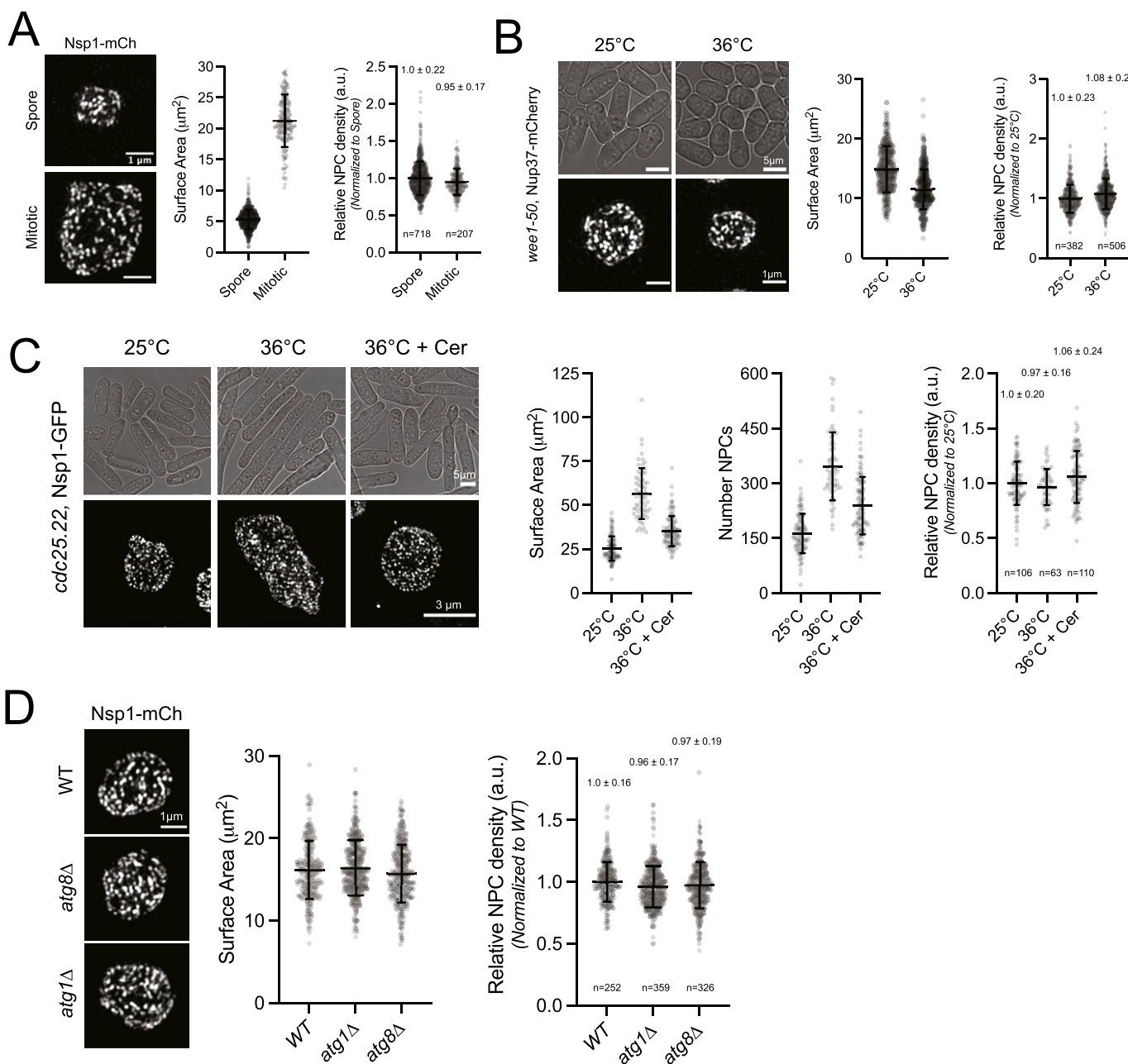

**Figure 2. Surface area-dependent maintenance of nuclear pore complex (NPC) density.**
**(A)** 3D-SIM image and quantitation (mean ± SD) of Nsp1-mCh nuclei from meiotic (top) or mitotic (bottom) nuclei. n, number nuclei. **(B)** 3D-SIM and quantitation of Nup37-mCherry NPCs in *wee1.50* mutants grown at 25°C or shifted to 36°C for 3.5 h. **(C)** 3D-SIM and quantitation of Nsp1-GFP in *cdc25.22* mutants at 25°C or shifted to 36°C for 3.5 h in the absence or presence of 10 *µ*M cerulenin (Cer). **(D)** 3D-SIM and quantitation of Nsp1-mCh NPCs in wild-type, *atg1Δ*, and *atg8Δ* cells. Bars, 1 *µm*.

two independent mechanisms exist to control NPC cluster dynamics and transmission during *S. pombe* nuclear division.

**Reduced NPC density and altered basket composition over the nucleolus**

We observed a clear reduction in NPC density over the nucleolus (visualized using the RNA polymerase I subunit Nuc1-mCh) (Hirano et al, 1989) from middle slices of 3D-SIM images (Fig 4A), suggesting that a similar reduction of NPC density over the nucleolus occurs in *S. pombe* like *S. cerevisiae* (Wang et al, 2016). To quantitatively assess NPC density over this region, we compared the intensities for multiple Nups from all subcomplexes over the nucleolus with intensities over the rest of the NE (Fig 4B and C). This confirmed that NPC density is reduced by ~20% over the nucleolus-facing region of the NE and showed a ~50% average reduction for the *S. pombe* Tpr

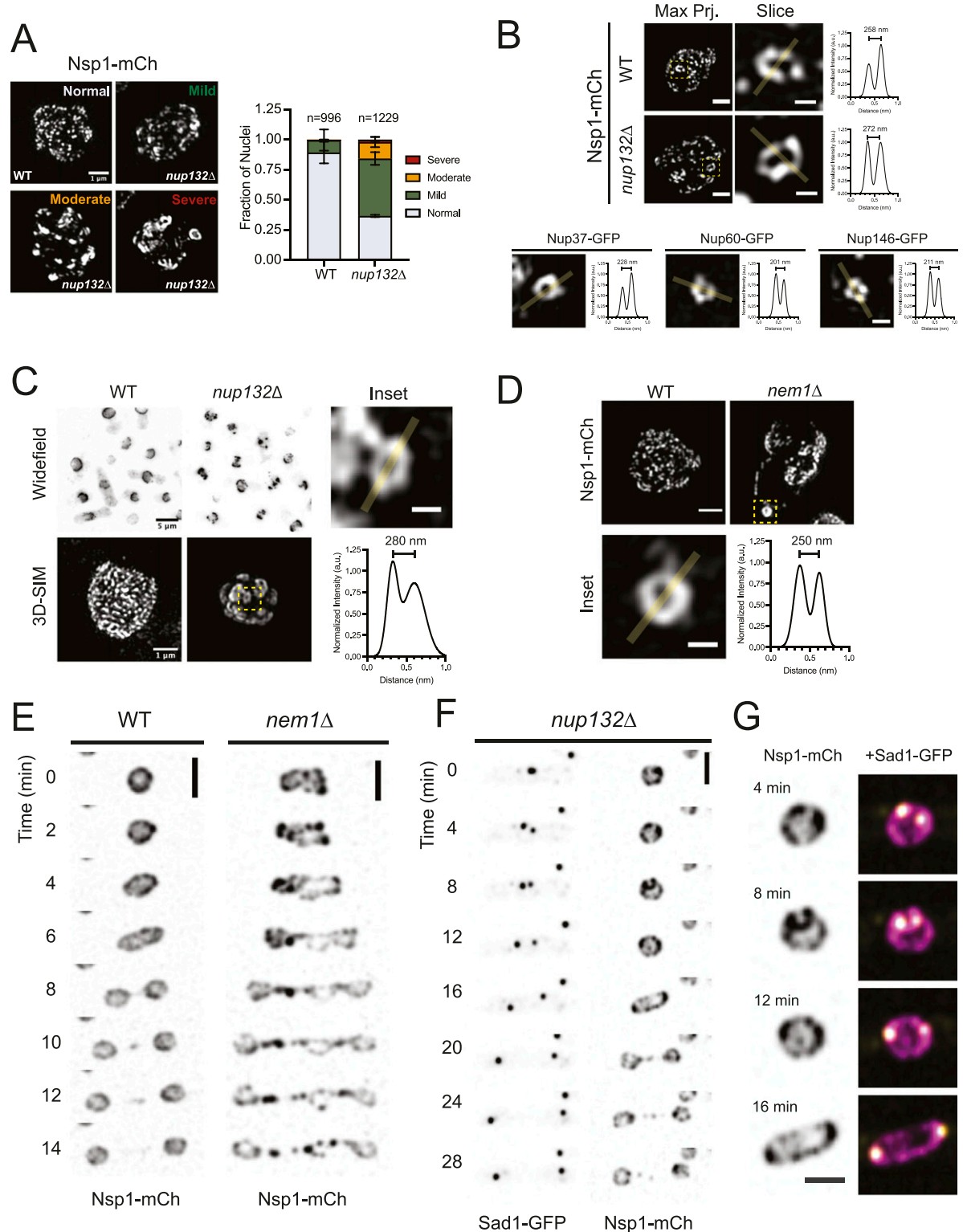

**Figure 3. Nuclear pore complex (NPC) cluster organization and dynamics.**
**(A)** 3D-SIM of Nsp1-mCh in wild-type (WT) and *nup132Δ* cells. Clustering frequency from two independent replicates shown at right. Bar, 1 μm. **(B)** Ring-like NPC clusters observed by 3D-SIM in projections of the entire nucleus (left; bar, 1 μm) and a subregion (center; bar, 300 nm) with intensity profiles of the highlighted region shown at right. Below, images of ring clusters in WT cells with Nups from multiple subcomplexes. **(C)** Clustering increases in *nup132Δ* cells grown on YES agar plates at 25°C for 7 d. **(D)** 3D-SIM of Nsp1-mCh NPCs in WT and *nem1Δ* mid-G2 stage nuclei (Bar, 1 μm), with ring cluster shown in the inset (Bar, 300 nm) and plot profile. **(E)** Montage of time-lapse images of Nsp1-mCh in WT and *nem1Δ* cells. Bar, 5 μm. **(F)** Montage of Nsp1-mCh and the spindle pole body component Sad1-GFP in *nup132Δ* mutants. Bar, 5 μm. **(F, G)** Insets of nuclei at the indicated time points from montage in (F). Bar, 3 μm; Nsp1, magenta; Sad1, yellow.

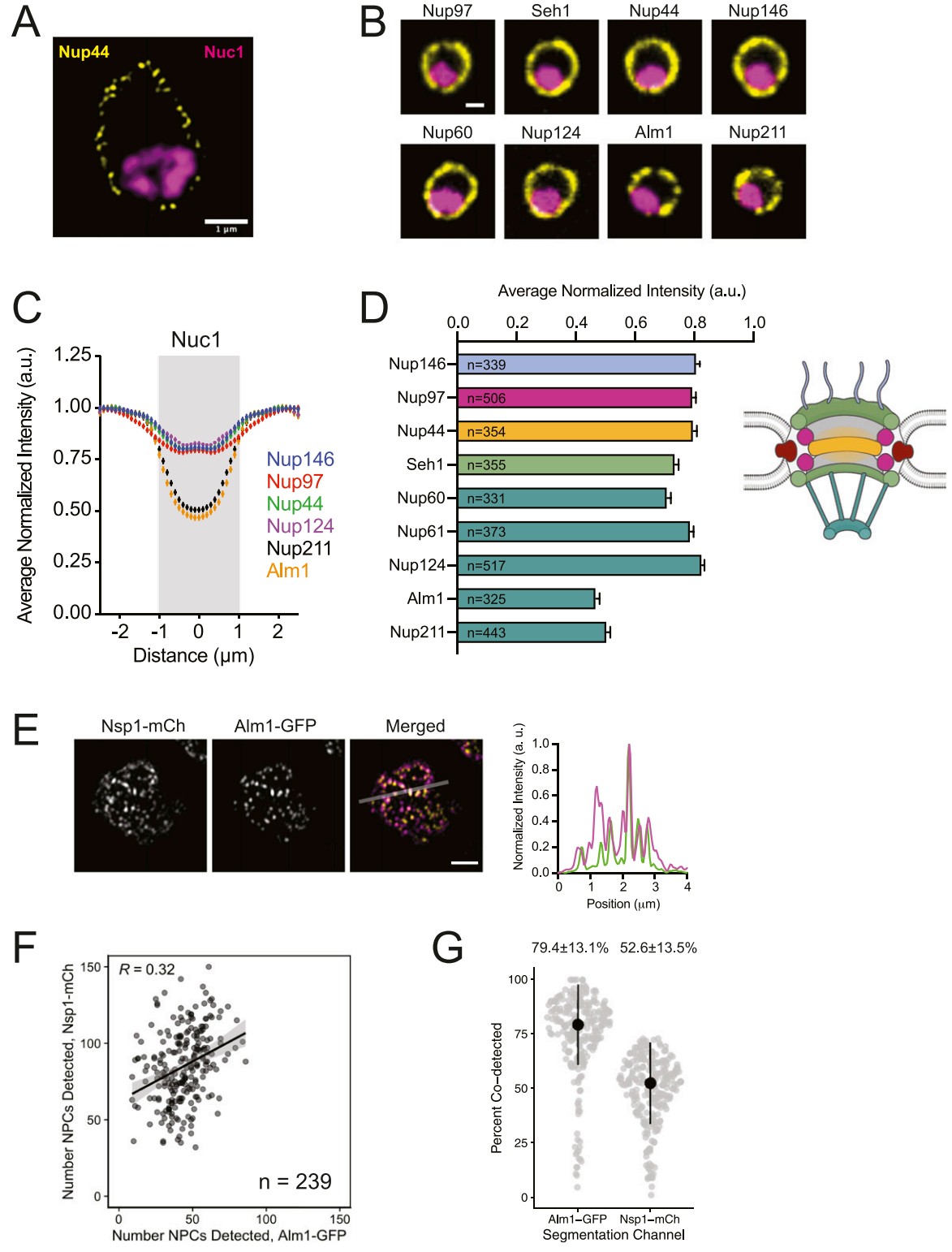

**Figure 4. Reduced nuclear pore complex (NPC) density and altered NPC composition at the nucleolus.**
**(A)** Middle slice from 3D-SIM image of Nup44-GFP NPCs and the nucleolus (Nuc1-mCh, processed with Gaussian blur). Bar, 1 µm. **(B)** Confocal images showing Nups (yellow) and the Nuc1 (magenta). Bar, 1 µm. **(C)** Averaged profiles of Nup intensities at the nuclear envelope relative to Nuc1 (gray, based on full width half maximum ±95% CI of Nuc1). **(C, D)** Average Nup intensity at position 0 in intensity profiles from (C), organized by NPC subcomplex. **(C, D)** Error bars, SD. N, number of nuclei analyzed for panels (C) and (D). **(E)** 3D-SIM image of Nsp1-mCh and Alm1-GFP with intensity profile of indicated region. **(F, G)** Comparison of the NPC number and fraction co-detected NPCs. The linear model regression line with 95% CI shown in (F), R, Pearson's coefficient; median and median absolute deviation indicated above each group in (G).

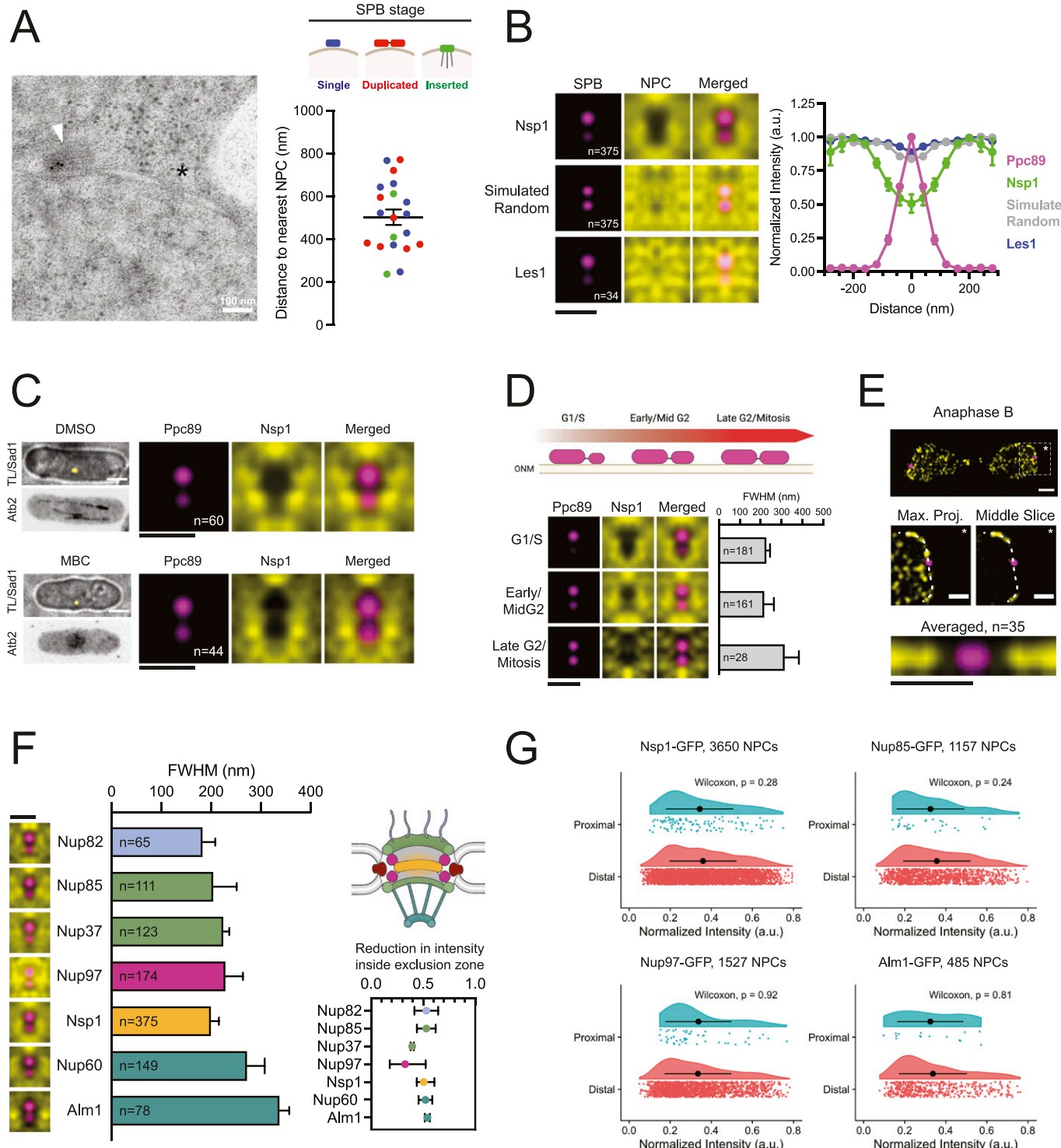

**Figure 5. Nuclear pore complex (NPC) exclusion from the spindle pole body (SPB) proximal region of the nuclear envelope.**
**(A)** Immuno-EM of SPB component Ppc1-GFP (arrowhead). The nearest NPC is highlighted with an asterisk. Plot of distance from SPB to the nearest NPC, based on SPB stage: blue = single SPB; red = duplicated SPB; green = inserted SPB. **(B)** SPA-SIM images of Ppc89-mCh (magenta) and Nsp1-GFP (NPCs), Les1-GFP and simulated random distributions (yellow). Normalized intensity profiles across the mother and daughter SPBs. Error bars, SD. Bar, 0.5 μm. **(C)** Confocal image for microtubules (mCh-Atb2) and SPBs (Sad1-GFP) in cells treated with DMSO (control) or 25 μg/ml MBC for 1 h. Bar, 3 μm. SPA-SIM images of Ppc89-mCh (magenta) and Nsp1-GFP (yellow) for cells similarly treated. Bar, 0.5 μm. **(D)** Schematic of SPB duplication. SPA-SIM images of Ppc89-mCh (magenta) and Nsp1-GFP (yellow) based on daughter/mother Ppc89-mCh intensity ratios (G1/S, 0.5; Early/Mid G2, 0.5–0.8; Late G2/Mitosis, ≥0.8). Plot of full width half maximum of Nsp1 exclusion zone for each stage. Error bars, 95% CI. Bar, 0.5 μm. **(E)** 3D-SIM projection of Nsp1-GFP (yellow) and Ppc89-mCh (magenta) in anaphase. Bar, 1 μm. Enlarged images of SPB region, showing maximum projection and single middle z-slice. Bar, 0.5 μm. Averaged image of Nsp1-GFP NPCs relative to SPB in mitotically dividing nuclei. Bar, 0.5 μm. **(D, F)** SPA-SIM images of Nups (yellow) and SPBs

orthologs Alm1 and Nup211 (Jiménez et al, 2000; Bae et al, 2009; Salas-Pino et al, 2017) (Fig 4D). Two-color SIM data suggest that a significant population of NPCs exist in *S. pombe* that lack the Alm1/Nup211 basket as ~50% of NPCs (visualized with Nsp1-mCh) lacked Alm1-GFP (Fig 4E–G). Together, these results show that like budding yeast, the NE region over the nucleolus has reduced NPC density and is enriched for a population of NPCs that specifically lack the Tpr basket nucleoporins in *S. pombe*.

### NPCs are excluded from the SPB proximal region throughout the mitotic cell cycle

In contrast to the nucleolar region, increased NPC density is found near the budding yeast SPB (Winey et al, 1997), possibly because of a role of NPCs in NE remodeling during SPB insertion into the NE (Rüthnick et al, 2017). EM analysis of fission yeast SPBs failed to identify an increased presence of NPCs within ~200 nm of the SPB regardless of cell cycle stage (Fig 5A). To examine the distribution of NPCs relative to the SPB at high resolution, we used single particle averaging of multiple 3D-SIM images (SPA-SIM); this approach has allowed us to visualize SPB-proximal proteins in budding and fission yeast (Burns et al, 2015; Bestul et al, 2017, 2021; Chen et al, 2019). In SPA-SIM, the position of the two duplicated but unseparated SPBs (visualized using the SPB marker Ppc89-mCherry) are used as fiduciary points to realign images from multiple nuclei. To ensure that NPC distribution is visualized from a top-down perspective, we restricted our SPA-SIM analysis to SPBs that were centrally localized in the x–y plane with respect to the nucleus (see the Materials and Methods section). A composite image was then generated, representing the average distribution of proteins of interest with respect to the SPBs.

Consistent with EM analysis of NPC distribution around the fission yeast SPB, we observed a clear zone of NPC exclusion surrounding the SPBs in asynchronous populations of exponentially growing cells (Fig 5B). This exclusion was not seen for Les1, an INM protein that is not a component of the NPC (Dey et al, 2020) or from simulations of randomly distributed NPCs (Fig 5B). This exclusion zone was highly reproducible and cell cycle–independent (Fig 5D and E), with an average diameter of ~200 nm (FWHM = 183.8–217.2 nm, 95% CI). We considered that SPB exclusion of NPCs could be because of forces exerted on SPBs during the interphase through the activity of microtubule-based motor proteins (Tran et al, 2001). However, the exclusion zone was not altered after disruption of microtubules by treatment with the depolymerizing agent methyl benzimidazol-2-yl carbamate (MBC) (Tran et al, 2001; Sawin & Snaith, 2004) (Fig 5C).

We envisioned at least two possible models that could explain the NPC exclusion near the SPB. In the first model, NPCs could be physically excluded from this region, perhaps through the presence of nuclear membrane proteins localized to the SPB region. This could include factors such as the SUN (Sad1-Unc-84 homology)

domain–containing protein Sad1, which interacts with KASH (Klarsicht, ANC-1, Syne Homology) domain proteins Kms1 and Kms2 to form a LINC (linker of nucleoskeleton and cytoskeleton) complex that tethers the cytosolic SPB to the NE (Miki et al, 2004; King et al, 2008) or through proteins that tether centromeres to the NE (Gallardo et al, 2019). Alternatively, the exclusion could represent a localized region of the NE containing NPCs with reduced Nup intensity, perhaps representing partial NPC disassembly or assembly intermediates.

Multiple lines of evidence suggest that the reduced Nup intensities near the SPBs are the result of physical exclusion of NPCs from this region. First, we observed similar exclusion patterns for multiple Nups including members of each NPC subcomplex (Fig 5F). The size of the exclusion zone was relatively stable for all Nups, although was slightly larger for components of the nuclear basket (Fig 5F). Most of the Nups were reduced by ~50% over the SPB proximal region, although the structural components Nup97 and Nup37 were excluded to a lesser extent (Fig 5F inset). If the observed Nup exclusion was because of changes in NPC composition near the SPB, we expected to see a relationship between NPC/Nup intensity and proximity to the SPBs. However, the intensities for individual NPCs that were proximal (within 100 nm radius of SPBs) and those that were distal were equivalent for multiple Nups and only marginally (~20%) reduced for a subset of Nups (Figs 5G and S3). Collectively, these findings support a model in which the reduced Nup intensity surrounding the SPBs is the result of reduced presence of NPCs in this region, rather than localized alterations of NPC composition.

### Exclusion of NPCs near the SPBs requires Lem2 and centromere tethering

We recently showed that the INM protein Lem2 localizes to the SPB during the interphase and forms a ring with similar dimensions to that of the NPC exclusion zone (Hiraoka et al, 2011; Bestul et al, 2021) (Fig 6A). We hypothesized that the Lem2 ring may be a component of the physical barrier that prevents NPCs from localizing to this region. Indeed, deletion of *lem2+* resulted in a significant decrease in NPC exclusion from the SPB region (Fig 6B) without affecting NPC composition as SPB proximal and distal Nsp1-GFP intensities were similar in *lem2Δ* mutants. A decrease in NPC exclusion was not seen in cells lacking the INM protein Ima1 or the second *S. pombe* LEM domain–containing protein Man1 that does not localize to the SPB (Hiraoka et al, 2011) (Fig 6B).

Lem2 contains two nucleoplasmic regions: an N-terminal HEH/LEM domain that is required for DNA binding and centromere tethering at the SPB (Barrales et al, 2016; Fernández-Álvarez et al, 2016) and a C-terminal Man1/winged-helix domain that tethers telomeres to the nuclear periphery (Gonzalez et al, 2012) (Fig 6C). Lem2 truncation mutants lacking the N- and C-termini localize to the SPB (Barrales et al, 2016), allowing us to test which regions of

(magenta), along with full width half maximum plot as in (D). Most of the Nups have ~50% reduction in intensity near the SPB relative to the surrounding nuclear envelope. Bar, 0.5 *μ*m. **(G)** Kernel-smoothed density distributions of Nup-GFP intensities for NPC foci that were proximal (<100 nm) or distal (>100 nm) to the SPB. Nup intensities for proximal and distal NPCs were compared using the unpaired Wilcoxon rank-sum test. Black dots represent the mean normalized intensity value, and error bars show SD.

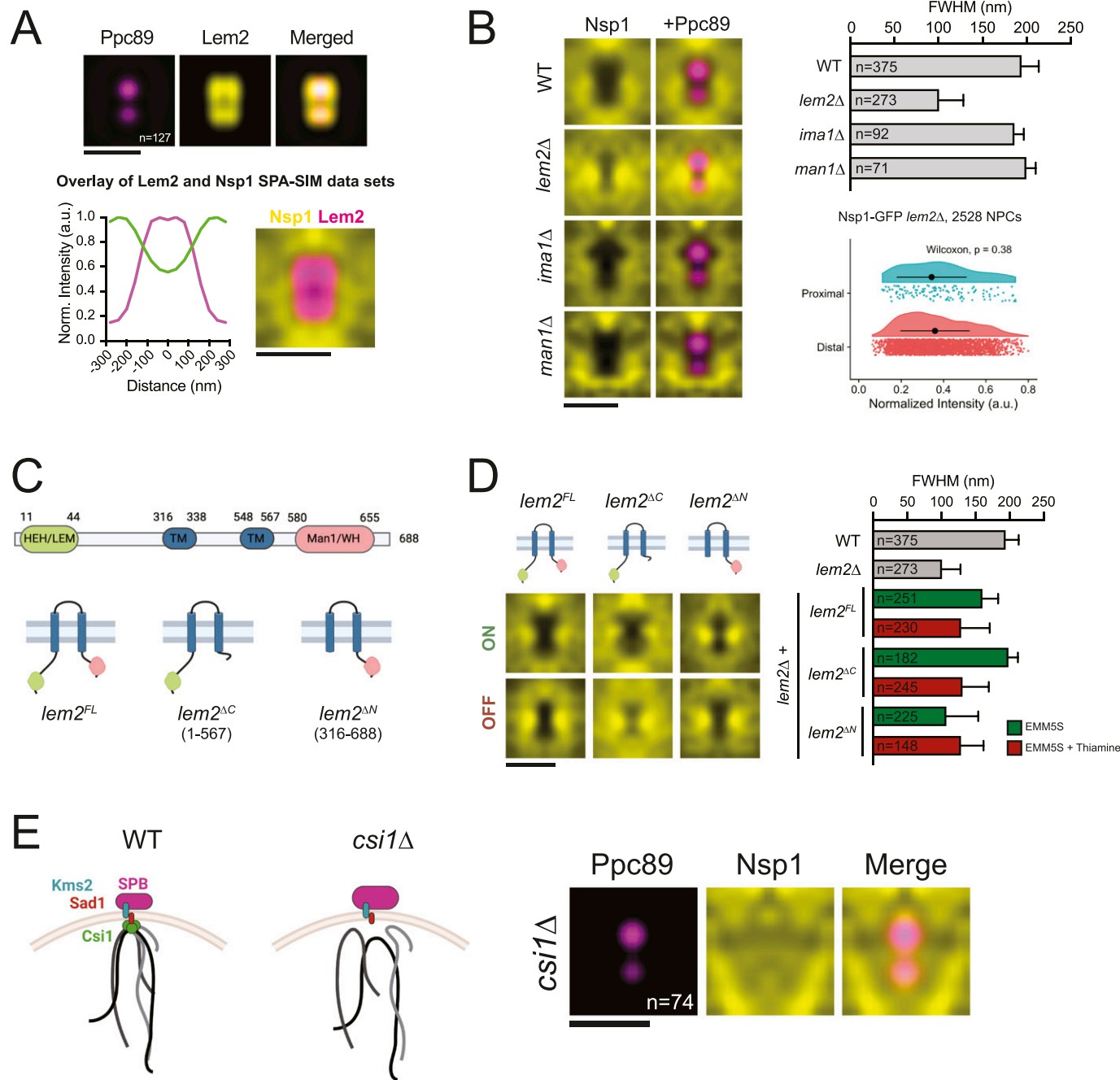

**Figure 6. Nuclear pore complex (NPC) exclusion requires Lem2 and centromere tethering.**
**(A)** SPA-SIM of Lem2-GFP (yellow) and Ppc89-mCh (magenta). N, number of averaged images. Lem2-GFP and Ppc89-mCh intensity profiles are shown below. Overlay of SPA-SIM datasets for Lem2-GFP (magenta) and Nsp1-GFP (yellow). Bars, 0.5 μm. **(B)** SPA-SIM Nsp1-GFP (yellow) and Ppc89-mCh (magenta) in wild-type, *lem2Δ*, *ima1Δ*, and *man1Δ* backgrounds. Bar, 0.5 μm. Plot of NPC exclusion zone dimensions (full width half maximum, ±95% CI). Nsp1-GFP intensity in spindle pole body proximal and distal regions was plotted and compared using a Wilcoxon rank-sum test. Black dots represent the mean normalized intensity value, and error bars show SD. **(C)** Schematic of Lem2, including rescue constructs. **(D)** SPA-SIM images of Nsp1-GFP (yellow) in *lem2Δ* cells with *lem2^FL^*, *lem2^ΔC^*, or *lem2^ΔN^* constructs turned on or off. Bar, 0.5 μm. Plot of NPC exclusion zone dimensions (full width half maximum, ± 95% CI). **(B)** For comparison, wild-type and *lem2Δ* dimensions from (B) are also shown. **(E)** Schematic of centromere tethering at the spindle pole body in wild-type and *csi1Δ* cells. SPA-SIM of Nsp1-GFP (yellow) and Ppc89-mCh (magenta) in *csi1Δ* cells. N, number of averaged images. Bar, 0.5 μm.

Lem2 are needed for NPC exclusion. Full-length or mutant versions of Lem2 were stably integrated at the *ura4+* locus and expressed as C-terminal 3xHA fusion proteins in a *lem2Δ* background using the thiamine-regulatable *nmt41+* promoter system (Basi et al, 1993;

Forsburg, 1993) (Figs 6D and S4). Exclusion of NPCs from the SPB region was similar to wild-type cells when either full-length (*lem2^FL^*) or *lem2^ΔC^* constructs were expressed (EMM5S). However, *lem2^ΔN^* expression resulted in an exclusion zone FWHM similar to *lem2Δ*

mutants (Fig 6D), suggesting that NPC exclusion depends on the function of Lem2's DNA-binding N-terminal HEH/LEM domain.

The size of the NPC exclusion zone was reduced in *lem2Δ* and *lem2^{ΔN}* strains; however, NPCs were still strongly excluded from a smaller region directly underneath the SPBs. During interphase, fission yeast centromeres tether under the SPBs (Funabiki et al, 1993) through interactions with multiple proteins including Lem2, Sad1, and Csi1 (reviewed in Gallardo et al [2019]). We hypothesized that the smaller exclusion zone observed in the absence of Lem2 could reflect a physical barrier formed by the remaining NE–centromere interactions. To test whether tethering of centromeres to the SPBs drives exclusion of NPCs from this smaller region, we examined NPC exclusion in *csi1Δ* mutants, in which ~70% of cells exhibit defects in centromere tethering (Hou et al, 2012). Interestingly, in *csi1Δ* cells, NPCs were no longer excluded from the SPB-proximal region (Fig 6E). This supports a model whereby the exclusion of NPCs from the SPB region is the result of physical interactions between centromeres and INM proteins, including Lem2, that tether the centromeres under the SPBs during interphase.

## Discussion

Multiple imaging approaches, including EM and fluorescence microscopy, have been used to determine the number and distribution of NPCs in various systems. The higher resolution afforded by EM and super-resolution light microscopy methods often comes at a price of significant increases in the time required for sample preparation, image acquisition, and analysis. In contrast, 3D-SIM generates high-resolution datasets using standard fluorescence microscopy approaches, allowing for quantitative analysis of NPC organization through whole nuclei. We apply 3D-SIM to fission yeast nuclei to provide the first map of NPCs in this system. We find that NPC density in *S. pombe* nuclei is similar to densities described for many metazoan nuclei and is constant over a range of nuclear sizes. The ~10% reduction in NPC densities seen in late mitotic and G1/S nuclei was not observed using measurements of total Nup-GFP intensity (Fig 1E). This could reflect cell cycle–specific alterations in NPC distribution that promotes formation of clusters that cannot be resolved by 3D-SIM, similar to the increase in clustering observed in *S. cerevisiae* nuclei during mitosis (Winey et al, 1997). However, the finding that NPC density is maintained through the *S. pombe* cell cycle has important implications regarding the mechanisms used for NPC assembly in fission yeast. For example, the total number of NPCs present in the two daughter nuclei in late mitosis is ~26% greater than the number present in the mother nucleus prior to division. In agreement with previous findings (Neumann & Nurse, 2007), we observed that although the combined nuclear volume of the daughter cells is similar to the total volume in the mother nucleus, the combined surface area is ~34% greater than that of the mother nucleus (Table S1). Together, this suggests that NPC assembly continues to occur during the rapid expansion of the NE during cell division (Lim et al, 2007). However, NE expansion during mitosis takes place over a time frame of roughly 20–25 min, significantly shorter than the 45–60 min required for completion of NPC assembly in budding yeast and during the interphase in metazoans (Otsuka et al, 2016; Onischenko et al, 2020). During the

short cell cycles of the syncytial nuclear divisions in *Drosophila* embyros, rapid NPC assembly occurs via incorporation of assembly intermediates from annulate lamellae (Hampoelz et al, 2019, 2016). However, by EM and by fluorescence microscopy, we and others do not observe pools of NPCs/Nups outside the NE so it is unclear how *S. pombe* maintains NPC density during mitosis. Continued NPC assembly in *cdc25.22* arrested cells that have low Cdk1 activity suggests that unlike in metazoans, Cdk1 (SpCdc2) is not required for NPC assembly in *S. pombe*. The fact that the nuclear size increase and NPC assembly in *cdc25.22* were blocked by inhibiting fatty acid synthesis supports a model for negative regulation of NPC assembly similar to that proposed in vertebrates mediated by Tpr/ERK. In this model, signals emanating from existing NPCs inhibit assembly of NPCs in the surrounding region and inhibition of NPC assembly can be overcome by reducing NPC density through NE expansion. The mechanistic details of this regulation likely differs between species as neither the MAPKs active during vegetative growth (Sty1 and Pmk1) (Shiozaki & Russell, 1995; Toda et al, 1996) nor the *S. pombe* ERK ortholog (Spk1) (Toda et al, 1991) are essential for cell viability.

A major benefit of the 3D-SIM approach is that the improved resolution allowed for identification of distinct patterns of organization for clustered NPCs in *S. pombe*. NPC clusters were often observed to be organized in ring-like patterns ~200–300 nm in diameter that were more prevalent in the clustering mutants *nup132Δ* and *nem1Δ*. These rings are smaller than typical yeast autophagosomes (300–900 nm diameter) (Takeshige et al, 1992; Baba et al, 1994), although they are similar in size to nuclear-derived vesicles containing NPCs seen in EM images of NPCs being removed by autophagy in budding yeast (Lee et al, 2020). Our observation of an increased ring number by 3D-SIM when *nup132Δ* cells were grown on solid instead of liquid media suggests that changes in nutrient availability or NE composition triggers NPC reorganization into ring clusters of a consistent size. Whether the formation of these rings promotes their subsequent removal via autophagy or other pathways remains to be tested. However, the increased frequency of ring clusters in *nup132Δ* cells may provide insights into the mechanism driving their formation. Nup132 is a structural Nup that facilitates interactions between the structural scaffold of the NPC with lipids through its N-terminal ALPS motif (Fernandez-Martinez et al, 2012; Nordeen et al, 2020). Deletion of Nup132 may alter the interactions between NPCs and specific lipid species present in the NE, making *nup132Δ* cells especially sensitive to changes in lipid composition that may occur because of nutrient availability during growth on plates. Increased clustering during nutrient depletion may also contribute to the observed defects in sporulation seen for *nup132Δ* cells (Asakawa et al, 2014).

Time-lapse imaging of NPC clusters revealed two strikingly different approaches to how clustered NPCs are handled during mitosis. In *nem1Δ* mutants, both excess nuclear membrane material and NPCs are segregated into the anaphase bridge region during nuclear division (Fig 3F), a distinct nuclear compartment that is a unique site of NPC disassembly (Dey et al, 2020; Expósito-Serrano et al, 2020). Clusters of NPCs formed in *tts1Δ* cells specifically during mitotic NE expansion also localized to the anaphase bridge (Zhang & Oliferenko, 2014). This suggests that the anaphase bridge region of the NE may serve as a site where NPCs and NE material are sent to

be removed during division, analogous to the NE-derived compartment that forms in budding yeast meiosis II to sequester and degrade NPCs (King et al, 2019; Koch et al, 2020). Further, although NPCs can be selectively removed via autophagy in response to nutrient depletion in *S. cerevisiae* (Lee et al, 2020; Tomioka et al, 2020), removal during mitosis by sequestration into the anaphase bridge is the only mechanism we have identified to control the number of NPCs downstream of NPC assembly in *S. pombe*. The fact that *nup132Δ* clusters do not similarly localize to the anaphase bridge suggests that the fate of NPC clusters depends on the mechanisms driving the clustering. If *nup132Δ* clusters interact with specific lipids, this may also explain their portioning with the SPB, which has been proposed to contain a unique NE composition (Jaspersen & Ghosh, 2012).

Our results clearly demonstrate that the region of the NE over the nucleolus and near the SPB is distinct from other NE regions. It is not surprising that the nucleolar region has reduced NPC density and pores lacking the basket Nups Alm1 and Nup211 (Fig 4), given similar observations of the NPC number and composition in plants, mammals, and fungi (La Cour & Wells, 1974; Severs et al, 1976; Maul, 1977). In contrast, we were somewhat surprised to see reduced NPC density at the SPB in fission yeast given that NPC density is increased near SPBs in budding yeast (Winey et al, 1997; Rüthnick et al, 2017). Perhaps, this is reflective of differences in the roles NPCs play in SPB insertion into the NE in the two fungi—NPCs are thought to facilitate SPB incorporation into the NE in *S. cerevisiae* but do not appear to be required for SPB assembly into the NE in *S. pombe*.

A key question that remains is how populations of distinct NPC composition are established and maintained in specific regions of the NE because fungal NPCs laterally diffuse through the NE. At least three potential models exist: intact NPCs diffuse into a subregion and are partially disassembled; a unique NPC subpopulation is assembled in that region of the NE; or subregions of the NE have unique properties and preferentially allow for NPCs of specific composition to diffuse in and/or be retained. Partial disassembly of fungal NPCs has been reported in multiple cell types and conditions (De Souza et al, 2004; Gallardo et al, 2020; Meinema et al, 2021 *Preprint*). However, with respect to the SPB-proximal region, we favor a model for physical NPC exclusion involving both centromeres and Lem2. In this model, tethering of centromeres to INM-localized SPB components forms a physical barrier that prevents the diffusion of NPCs through the NE into the SPB proximal region. Recent work identified that Nups, including the basket Nup211, specifically interact with pericentromeric heterochromatin regions, whereas Lem2 interacts with the central core region of the centromere (Iglesias et al, 2020). The reduced presence of NPCs in the SPB proximal region likely reflects a functional consequence of these patterns of heterochromatin binding. Consistent with the steric model, if we reduced or eliminated centromere tethering, either by removing Lem2's N-terminal HEH/LEM domain or by deletion of *csi1+*, the NPC exclusion zone was diminished. Interestingly, NPCs remain excluded from the SPB region throughout mitosis, including during periods where Lem2 no longer localizes to the SPB (Hiraoka et al, 2011). During these stages, NPC exclusion is likely maintained by multiple proteins that form SPB-ring structures during mitosis, including Ima1 and Sad1 (Bestul et al, 2021).

It is likely that the reduced NPC density and altered basket composition over the nucleolus is produced through a different mechanism. In budding yeast, the NE over the nucleolus is more amenable to membrane expansion than regions outside of the nucleolus (Campbell et al, 2006). Similar differences in NE membrane properties over the nucleolus may exist in *S. pombe* and could drive the observed NPC heterogeneity. For example, differences in membrane composition or fluidity could alter the ability for NPCs to diffuse laterally through this portion of the NE, leading to reduced density over the nucleolus. Alternatively, the region over the nucleolus could have higher rates of NE membrane incorporation and NPC assembly. In this scenario, the reduced presence of Alm1 and Nup211 could be because of these Nups being the last components added during NPC assembly (Onischenko et al, 2020). In either case, the reduction in Alm1/Nup211 over the nucleolus could be the result of interactions between chromatin and NPCs containing Alm1/Nup211 (either directly or indirectly via basket-associated complexes involved in mRNA processing and export) that may prevent their diffusion back into the nucleolar-facing NE compartment. Our results establish *S. pombe* as a model for further studies determining the mechanisms that establish and maintain distinct populations of heterogeneous NPC composition within single nuclei. Importantly, our results demonstrate that the reduced NPC density and specific loss of Tpr-ortholog basket components over the nucleolus is not unique to budding yeast but is a conserved feature of nuclear organization across highly divergent species. The ability for 3D-SIM to resolve and quantify individual NPCs labeled with multiple fluorescent proteins at endogenous levels provides tools to begin to interrogate how altered NPC compositions may allow for functional specialization of NPC function at distinct regions of the NE.

## Materials and Methods

### Yeast strains and plasmids

All *S. pombe* strains used in this manuscript are listed in Table S2. Deletion strains were obtained from the *S. pombe* haploid deletion library (Bioneer). Genes of interest were endogenously tagged using standard PCR-based methods (Bähler et al, 1998), with lithium acetate transformation and colony selection as previously described (Murray et al, 2016). Cells were cultured in yeast extract with supplements (YES) media (5 g yeast extract, 30 g dextrose, 0.2 g each adenine, uracil, histidine, leucine, and lysine, in 1 liter of water) at 25°C, unless otherwise noted. For experiments using the *nmt41+* promoter, cells were cultured in Edinburgh minimal media with amino acid supplements (EMM5S) (Petersen & Russell, 2016) at 30°C. Thiamine was added to EMM5S to a final concentration of 15 $\mu$M for 18–24 h at 30°C to repress expression. All strains were maintained in liquid culture for at least 48 h with back diluting to maintain cultures in logarithmic growth before imaging, unless otherwise noted. Where noted, cultures were treated with methyl benzimidazol-2-yl carbamate (MBC, 25 $\mu$g/ml), cerulenin (10 $\mu$M), or dimethylsulfoxide (DMSO, vehicle control).

The coding sequence, or subdomain regions, for *lem2+* was amplified from genomic DNA using HiFi PCR master mix (Clontech)

and cloned into NdeI/XhoI–digested pREP41-MCS+. The resulting plasmid was used as a template to amplify *nmt41-lem2-3xHA*, which was transformed into the *ura4+* locus of *lem2Δ* cells as described (Vještica et al, 2020). Lem2 mutants were similarly cloned and integrated. Integration was verified by PCR, and thiamine-dependent repression was validated by Western blotting of whole cell extracts using anti-HA antibodies (3F10; Roche).

**NPC quantitation and analysis by 3D-SIM**

Exponentially growing cells were collected by centrifugation for 3 min at 3,000 rcf and fixed in a solution of 4% formaldehyde supplemented with 200 mM glucose. Fixed cells were imaged in phosphate buffered saline, pH 7.4, with an Applied Precision OMX Blaze V4 (GE Healthcare) equipped with a 60× 1.42 NA Olympus Plan Apo oil objective and two PCO Edge sCMOS cameras. Two-color (GFP/mCherry) imaging was performed using 488-nm (GFP) or 561-nm (mCherry) lasers with alternating excitation and a 405/488/561/640 dichroic with 504–552-nm and 590–628-nm emission filters. Images were acquired over a volume covering the entire nucleus with z-spacing of 125-nm (typically 4 μm). Widefield images of NPC clusters were acquired using the same microscope and settings but operating in widefield mode. SIM images were reconstructed with Softworx (Applied Precision Ltd), with a Wiener filter of 0.001. Except where noted, SIM images shown throughout are maximum intensity projections of all z-slices, scaled using bilinear interpolation with linear brightness and contrast adjustments in ImageJ (Schneider et al, 2012).

Image analysis was performed using a number of custom plugins and macros for ImageJ, all of which are freely available at http://research.stowers.org/imagejplugins/. Additional documentation and source code used for NPC density analysis can be found at http://www.stowers.org/research/publications/libpb-1640. Statistical analysis was performed using R or GraphPad Prism v 9.0. Average values along with SD from the mean are shown based on the indicated number of nuclei analyzed (n), unless otherwise noted.

To quantitate the number of NPCs, individual nuclei were detected and segmented in an automated fashion using custom ImageJ plugins. Briefly, maximum intensity projections were used to perform automatic local thresholding for nuclear segmentation using a semiautomated protocol allowing the user to add and remove missed or poorly segmented ROIs. Each nucleus was cropped, and NPCs were detected using a "track max not mask" approach, in which the brightest voxel in the image is found and a spheroid with a diameter of 8 pixels (320 nm) in x and y and 5 slices in z (625 nm) is masked around that voxel. This process repeats until no voxels remain above a minimum threshold of 25% of the maximum intensity in the image. After NPC detection, the three-dimensional coordinates were used to model the NE surface using the "*convhulln*" function from the *geometry* package in R. Occasionally, we observed the presence of points detected away from the NE (representing noise or foci of cytoplasmic signal). To remove these points before computation of the convex hull, we included an optimization step in which up to 10 percent of the initial points could be removed if doing so increased the fraction of points present on the convex hull surface. The surface area and volume metrics were extracted for the 3D convex hull and used to derive NPC density values.

To quantify the fraction of co-detection for two-color 3D-SIM images, NPCs were detected in both channels as described above. The coordinates of the points detected in each channel were used to determine whether a corresponding point was detected in the second channel by computing a distance matrix between all point pairs using the "*dist.xyz*" function in the R package bio3D (Grant et al, 2006). Points were considered to be co-detected if foci were found within a 250 nm radius.

For a secondary method to validate density measurements, the unprocessed, nonreconstructed image files were sum-projected and background subtracted, and the integrated intensities were measured for each nucleus using the ROIs generated during segmentation of nuclei for NPC quantitation as described above. For each nucleus, the integrated intensity was divided by the calculated nuclear surface area to derive a measurement for Nup intensity per unity surface area.

Cells were sorted into cell cycle stages using the following criteria: early G2, cell length < 9.5 μm, mononucleate; mid-G2, length between 9.5 and 11 μm, mononucleate; late G2/early mitosis, length ≥ 11.0 μm, mononucleate; late mitosis, length ≥ 11 μm, binucleate; G1/S, septated.

Single particle analysis was performed as previously described (Bestul et al, 2017). Briefly, mother and daughter SPB spots were manually selected, and each spot was fitted to two 3D Gaussian functions and realigned along the axis between these two functions. To allow for visualization of NPC distributions in the x–y plane relative to the SPBs, a Euclidean distance filter was used in ImageJ to select for images in which both SPB points were least 400 nm away from the edge of the nucleus based on maximum intensity projections. Realigned images were averaged as described previously (Burns et al, 2015; Bestul et al, 2017). To account for biologically irrelevant directional bias in the x–y plane, the averaged images were further averaged with a mirrored (x-axis) image. All averaged images presented were thresholded to display pixels above a threshold of 25% of the maximum intensity value. Quantitation of Nup intensity relative to the mother and daughter SPB was performed in an automated manner, using line profiles with a width (12 pixels) covering both mother and daughter SPBs to generate profiles of the Nup/GFP intensities. Proteins of interest were considered to be excluded if the normalized peak intensity fell below 0.8, based on the reduction in signal observed in simulated random datasets. For excluded proteins, the plot profiles were fit to a Gaussian and the width of the exclusion zone was measured by computing the FWHM value for the fit curve using the equation FWHM = 2.355 × SD. The size of exclusion zones for proteins of interest were compared using the FWHM value from plot profiles of the averaged SPA-SIM images, with error bars representing the 95% confidence interval of the SD of the Gaussian fit. Curve fitting and statistical analyses were performed using GraphPad Prism v. 9.0.

An analogous approach was used to generate an averaged image for Nups relative to the SPB in anaphase and telophase nuclei. First, nuclei in these stages were identified in which the SPB was clearly observed within the NE from single z-slices. These slices were then used to manually trace the NE (based on Nup signal), followed by straightening of the polyline using the ImageJ macro "polyline

profile jru v1" with a line width of 8 pixels and the option for "Output Straightened" selected. The resulting straightened images were used to identify the location of the SPB, and a 1-$\mu$m region of the image centered on the SPB was cropped. The cropped images were then combined, and an averaged image was generated. The averaged image was further averaged with a mirrored image (in both horizontal and vertical directions) to generate a final averaged image with no directional biases.

### Simulations and modeling

For comparison of SPA-SIM data to the distribution expected to be observed by random chance, we simulated spherical nuclei with a of radius 1.25 $\mu$m (based on average dimensions of mid-G2 stage nuclei) with 125 randomly positioned NPCs to model an NPC density (6.4 NPCs/$\mu$m$^2$) similar to the average NPC density observed by 3D-SIM. NPCs were simulated as 3D Gaussians with a FWHM in the x–y plane of 100 and 300 nm in z, a minimum center-to-center distance of 100 nm and a maximum intensity of 100 photons. The simulated pixel size was 40 nm with a z-slice spacing of 125 nm. The resulting intensities were multiplied by 20 (artificial "Gain"), and a Gaussian read noise with a SD of 40 intensity units was added to each voxel. For simulation of SPA-SIM data, two SPB points were simulated with a center-to-center distance of 180 nm in a second channel. Simulated images were processed using the same approaches outlined above for NPC quantitation and SPA-SIM analyses.

To estimate the accuracy of NPC quantification, simulations of randomly distributed NPCs were performed as described above for SPA-SIM but without inclusion of the SPB points. A range of simulated sphere sizes (radii from 600 to 1,800 nm, or 4.52 to 40.7 $\mu$m$^2$ surface area) and NPC densities (1–15 NPCs/$\mu$m$^2$) covering the ranges observed in our 3D-SIM experiments were performed, with 100 simulated images per condition. The simulated images were subjected to the NPC quantitation as described above, and the resulting values for NPC number, density, and nuclear surface area were compared with the known simulated values.

### Confocal imaging

Confocal imaging to determine NPC density over the nucleolus was performed in log-phase cells expressing the nucleolar protein Nuc1-mCherry and utilized a PerkinElmer UltraVIEW VoX with a Yokogawa CSU-X1 spinning disk head, a 100× 1.46 NA Olympus Plan Apo oil objective, and CCD (ORCA-R2) and EMCCD (C9100-13) cameras. GFP/mCherry images were taken using a 488-nm laser (for GFP/mNeonGreen) or a 561-nm laser (for mCherry), with alternating excitation. Images were collected using the Volocity imaging software with a z spacing of 0.3 $\mu$m over a volume of 8 $\mu$m. To assess nucleoporin intensity levels at the NE, the middle four slices of the image stack were sum projected using ImageJ, and the NE was manually traced to generate line profiles for both GFP (Nup) and mCherry (nucleolus) channels in nuclei where the nucleolus was oriented to one side of the nucleus in the x–y direction. The line profiles for Nuc1-mCherry were boxcar smoothened, thresholded at their half-maximal values, and all profiles were aligned at the center of the Nuc1-mCherry peak. The nucleoporin line profiles were then resampled and averaged to generate average intensity

profiles. Mean normalized intensity values for the Nup signal at the center of the nucleolar peak were calculated from three independent biological replicates and plotted in GraphPad Prism v. 9.0.

To analyze NPC cluster dynamics in live cells, ~200–300 $\mu$l of log phase cells were applied to 35-mm glass bottom dishes (no. 1.5 coverslip; MaTek) that had been pre-coated with 1 mg/ml soybean lectin (in water) for 15 min and rinsed with YES media. After cells were allowed to settle for 30 min at 30°C, 2 ml of pre-warmed YES media was carefully added. Cells were imaged on a Nikon Ti-E microscope equipped with a CSI W1 spinning disk (Yokogawa) using a 60× 1.4 NA Olympus Plan Apo oil objective and an iXon DU897 Ultra EMCCD (Andor) camera. GFP and mCherry were excited at 488 and 561 nm, respectively, and collected through ET525/36m (GFP) or ET605/70m (mCherry) bandpass filters. Samples were maintained at 30°C using an Oko Lab stage top incubator. Images were acquired over a 6-$\mu$m volume with 0.3-$\mu$m z-spacing for 45 min at 2 min intervals.

To generate image montages presented in Fig 3E and F, maximum intensity projections of the full image stacks were background subtracted, bleach corrected (using the Simple Ratio method), and scaled threefold (x and y) using bilinear interpolation.

### EM

The distance between SPBs and the nearest NPC was measured in images from samples prepared as previously described (Bestul et al, 2017). Sections in which both the outer and INMs were clearly resolved were used for analysis. The distance from the center of the SPB to the nearest NPC (determined based on visible fusion of the INM and ONM) was measured by manually tracing the NE using the polyline tool in ImageJ, and the data were plotted in GraphPad Prism v. 9.0.

## Data Availability

Original data and all source code underlying this manuscript can be downloaded from the Stowers Original Data Repository at http://www.stowers.org/research/publications/libpb-1640.

## Supplementary Information

## Acknowledgments

We thank Zulin Yu and members of the Stowers Microscopy Core Facility for imaging assistance. We thank Jennifer Gerton and Brian Slaughter for mentoring and feedback and members of the Gerton and Jaspersen labs for discussion and feedback on this manuscript. Schematics throughout were created using http://BioRender.com. Research reported in this publication was supported by the Stowers Institute for Medical Research and NIH-NIGMS under award number R01GM121443 (to SL Jaspersen). JM Varberg is a recipient of a Ruth L Kirschstein NRSA Postdoctoral Fellowship (F32GM133096).

## Author Contributions

JM Varberg: conceptualization, data curation, software, formal analysis, funding acquisition, validation, investigation, visualization, methodology, and writing—original draft, review, and editing.
JR Unruh: conceptualization, software, formal analysis, and writing—review, and editing.
AJ Bestul: investigation.
AA Khan: investigation.
SL Jaspersen: conceptualization, supervision, funding acquisition, writing—original draft, and project administration.

## Conflict of Interest Statement

The authors declare that they have no conflict of interest.

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
