## [Reviewer comments · Life Science Alliance]

Life Science Alliance

Quantitative analysis of nuclear pore complex organization in *Schizosaccharomyces pombe*

Joseph Varberg, Jay Unruh, Andrew Bestul, Azqa Khan, and Sue Jaspersen

DOI: <https://doi.org/10.26508/lsa.202201423>

Corresponding author(s): Joseph Varberg, Stowers Institute for Medical Research and Sue Jaspersen,

Review Timeline:	Submission Date:	2022-02-23
	Editorial Decision:	2022-02-24
	Revision Received:	2022-03-03
	Editorial Decision:	2022-03-04
	Revision Received:	2022-03-11
	Accepted:	2022-03-14

Transaction Report:

Please note that the manuscript was previously reviewed at another journal and the reports were taken into account in the decision-making process at *Life Science Alliance*.

Reviewer #1 Review

Comments to the Authors (Required):

In this revised version, the authors have followed the suggestions of the reviewers to clarify their previous manuscript, and took in account my minor comments. In contrast, the additional experiments performed do not address my major points in a satisfying manner. Hence, the resulting publication now mainly provides a properly-performed description of NPC distribution in *S. pombe*.

1) I appreciate that the authors made the effort to perform the critical control required for the tethering experiment (answer to my major point 1). Unfortunately, this control experiment demonstrated that their tethering approach was not appropriate; hence the authors have removed the Figure 7 of the previous ms. They indicate that they "hope to revisit this approach in future work after optimization of the tethering system". Therefore, the revised manuscript becomes even more descriptive than the previous version.

Indeed, there is no more data demonstrating, as still stated by mistake in the abstract of the revised version, that NPC exclusion around the SPB... is important for timely mitotic progression.

While removing this set of potential functional data, the authors do also not perform one simple experiment suggested by reviewer 2 (in his main point 3), that would have been to analyse whether Lem2 influences NPC distribution at the nucleolar region as it does underneath the SPB. Although the authors had all tools in hand to address this question, this was considered by the authors to extend beyond the context of their current manuscript.

Yet, this would perhaps have helped to determine the extent to which centromeres as such, rather than general chromatin crowding, influence local NPC distribution. In that case, the conclusion of their abstract that "interaction between the centromere and the NE" influences local NPC distribution should be (or not) modified towards "interaction between chromatin and the NE" influences local NPC distribution".

2) Regarding the NPC quantification methodology (main points 2 and 3)

In the revised version, the authors now better clarify the limit of their quantitative "NPC number" analysis.

- To estimate the extent of NPC undercounting, the authors had simulated randomly distributed NPCs, and show that in the range of NPC density they analyze in *S. pombe*, [but also of nuclear size as included in this revised version] the underestimation "fell within 10-30% of the true simulated value".

However, it becomes even clearer in the revised version (see the statistics of the last two panels of Figure 1E) that NPC

undercounting can also be caused by the non-random distribution/clustering of NPCs that could not be taken in account in their model.

As such, the authors apparently rather trust in figure 1E the conclusions of their alternative quantitative approach, namely density/ unit surface measurements that indicated no significant difference between the NPC G1/S cells and early to late G2 cells; while the difference is very significant ($p < 0.01$, **, or perhaps $< 0.005 = ***$) when NPC "numbers"/surface is measured. To my point, this partly dismisses the interest of this sophisticated "individual" NPC approach the manuscript initially focused on.

2b) By performing additional experiments, the authors confirm and better illustrate "the variations in NPC number detected using different tagged Nups" (raised in my main point 2b). However, they do not address the question of whether these differences could be caused by a mild clustering phenotype (apparent changes in density) or by a possible heterogeneity in NPC composition (as for instance observed in an extreme manner for the basket protein Alm1 _ new panels 4E-G). Indeed, the authors choose to perform dual color analysis using a novel set of Nups (Nup40-mCh and Nup44-GFP). Dual imaging of Nup97 or Nup60 (average density 4 NPC/ μm^2) versus Nsp1 (average density 6 NPC/ μm^2) would have been more pertinent.

Minor point:

I do not think that normalizing the NPC density plot in Figure 2 within each individual experiment/panel "avoids any potential confusion". Rather, it partially masks the variability caused by Nups tagging. Note also that if normalized, the value plotted should be in arbitrary units and not NPC number/ μm^2 .

Reviewer #2 Review

Comments to the Authors (Required):

With this revision, the authors have addressed all my questions and concerns and I believe that the manuscript has been improved; therefore I recommend publication as it is.

Reviewer #1 Review

Comments to the Authors (Required):

In this manuscript, the authors have characterized NPC number and distribution in fission yeast using SIM imaging and various GFP or mCherry-tagged Nups. Their major observations are that:

- NPC density is constant throughout the cell cycle and in mutants/growth conditions affecting nuclear size (Figs 1 and 2)
- two previously -described NPC clustering mutants differentially segregate their clusters in mitosis. Mainly in clustering mutants, but also in wt cells, some NPC clusters organize in ring-like structures (Fig 3)
- there is a decrease in NPC density and altered NPC basket composition (lack of the TPR/Mlp orthologs) in NE areas over the nucleolus (Fig 4);
- Throughout the cell cycle, NPCs are excluded from an ~200nm-wide area around the SPB (Fig 5).

*. Exclusion involves the N-terminal LEM domain of Lem2 (known to form rings around the central SBP structure) and is released in a *csi1Δ* cells in which centromere tethering to SPB is altered (Fig 6).

* Ectopic targeting Nsp1 to this region affects mitotic progression (Fig 7).

Together, this represents a well-performed and extensive description of NPC distribution in fission yeast. However, as detailed below, while varied aspects are covered, this study stays short at defining the mechanisms underlying these observations. Data regarding constant NPC density, NPC clustering and NPC basket alteration in the vicinity of the nucleolus mainly comfort previous studies in budding yeast. In addition, the outcome of some experiments should be more carefully analyzed before drawing definitive conclusions.

Major points:

1) My major scientific concern is related to the final experiment (Figure 7) that leads the authors to conclude that maintaining the NPC exclusion zone around the SPB is important for timely mitotic progression.

- Here, the authors forced the localization of Nsp1-GFP to the SPB by overexpressing the INM protein Bqt1 fused to GBP (GFP-binding domain), a reporter previously used to tether centromeres to the *S. pombe* SPB (via Mis6-GFP, Fernández-Álvarez et al., *Dev. Cell* 2016; Bestul, Yu, Unruh, ... and Jaspersen, 2021).

1a) A major drawback is that the authors assume that by tethering Nsp1 to the SPB, they will tether the entire NPCs. However, the NPCs are here only visualized via Nsp1 itself. Therefore, one cannot exclude that the mitotic phenotypes observed could merely be caused by the ectopic recruitment of Nsp1 to SPB. This may in turn affect either SPB function (for instance, Nsp1

features unfolded FG repeats, prone to phase separation, and interacting also with Kaps and Ran,...), or affect transport function by partially depleting endogenous Nsp1 from the NPCs, in turn indirectly affecting cell cycle progression.

1b) I am surprised by the choice of Nsp1-GFP. Indeed, the shared central channel and nuclear basket localization of *S. cerevisiae* Nsp1 mentioned by the authors was observed several years ago by immuno-EM (Fahrenkrog et al, 1998), using a Prot-A-tagged Nsp1-Cterm fusion, and was not subsequently confirmed by others teams in budding yeast (see for instance Kim et al., 2018; doi:10.1038/nature26003), nor to my knowledge in metazoans. Instead, it seems now accepted that at least in budding yeast and vertebrates, Nsp1 is a shared component of the central channel and cytoplasmic filaments (note that the scheme in Fig 1B depicting Nsp1 might be modified accordingly). More recently, immunoEM of spNsp1-GFP cells revealed that "The *S. pombe* homolog of the conserved Nup Nsp1 (scNsp1/hsNup62) was localized frequently in the cytoplasmic side and infrequently in the nuclear side of the NPC" (Asakawa et al., PLOS Genetics, 2019). Hence, the model in Fig 7A may not be appropriate. Would a (possibly dynamic) central channel Nup be expected to relocalize NPCs?

To address these two issues, the authors should :

- provide individual data points and kernel-smoothed density distribution of Nup-GFP intensities (as in Figs 5G) in the presence/absence of thiamine, to possibly discriminate between a diffuse recruitment of individual GFP-Nsp1 molecules and the recruitment of entire NPCs.
- demonstrate that indeed, Nsp1-GFP is able to tether the NPC to the SPB. As mCherry and GFP are already used in this assay, the authors may employ IF with HA/FLAG-tagged Nups and appropriate anti-Tag and secondary antibodies or with mAb414 (using as tether a non-FG-Nup) or alternatively, perform EM analyses as in fig 5A.
- test tethering via other bona-fide basket Nups (Nup60, Alm1, Nup211), and as negative control, a cytoplasmic Nup bearing or not FG repeats (Nup82 or Nup146)
- assess the effect in this assay of tethering Nsp1-GFP deletion mutants, either lacking the C-term NPC tether, or the FG-repeats.

For the latter two experiments, a first readout could be the impact on cell growth in the presence or absence of thiamine.

1c) a minor issue is that both Ppc89 and Bqt1-GFP are mCherry-tagged. Yet, there does not seem to be any change in the signal observed in the mCherry channel upon Thiamine addition, suggesting that the major signal is due to Ppc89-mCherry. Expressing the inducible Bqt1-mCh-GFP in a cell line solely expressing a GFP-tagged Nup would allow to ensure its proper expression and localization (is its localization as restricted as Ppc89, or forming rings as reported for Lem2 or Sad1 ?).

1d) Since the average intensity of the Nups is only reduced by 2-fold in the SPB "exclusion zone", some NPCs (2-fold less than in other NE areas) are normally present in the vicinity of the SPB. May this NPC-depleted zone simply reflect steric hindrance caused by the centromeres and would have no specific function?

2) regarding the NPC quantification methodology.

The first part of the manuscript builds up on the detection and quantification of individual NPCs. However, several points should be clarified to determine the extent to which these values should be considered as relative or absolute values.

In addition, the extent to which minor changes in NPC density could be detected should be clarified.

2a) "Correcting for the undercounting observed for our 3D-SIM approach" (Line 164).

Was this correction performed for all presented NPC quantification experiments (graphs and table S1) or only used for values provided in the main text ?

How was it performed? I assume based on the graphs in fig S1C that correspond to simulated 3D-SIM. However, in the methods section the description of simulations (Page 28) only refers to SPA-SIM data. Moreover, in that case, only one nuclear size was used for simulation [I assume, with a radius of 1.2 μm leading to a theoretical surface of 18 μm^2 as G2 cells rather than 1.25 μm as indicated on line 585]. Were simulations performed with distinct nuclear sizes ? Did it give similar outputs when applied to much smaller (late M and G1/S, 15 μm^2 ; wee1 cells at 36{degree sign}C? 11 μm^2 ...) or larger nuclei (25-50 μm^2 for cdc25.22 cells).

This aspect is critical to clarify, as this may cause changes in NPC numbers, possibly affecting some conclusions.

2b) On line 166, the authors estimate that the average density on fission yeast nucleus is 6.3-7.4 NPC/ μm^2 . However, as indicated by the authors, the NPC quantifications reveal variations in-between cell lines: averages values for Nsp1-mCherry [possibly before correction, see above] is 5.1 to 6.4 NPC/ μm^2 (as indicated in the text), but only 4.1 for Nup37-GFP and about 4.5 for Nup97-GFP (Fig S1).

Such a cell-line dependent variations may reflect a mild impact of GFP-tagging of some Nups on NPC assembly or stability. If this were the case, and assuming all NPCs are detected, this would provide a situation in which NPC density would be mis-regulated. To address this issue, the authors should determine if the low density observed for some cell lines (i.e., Nup37-GFP and Nup97-GFP) is reproducible.

2c) The large variability observed in-between cell lines and even in-between experiments for a given cell line, also raises the issue of the possible mis-detection of some NPCs. In other words, once correction has been applied, which fraction of the NPCs

is detected ?

To try and address this issue, the authors could count the number of NPCs present in double-labelled cell lines (one would expect values to be nearly identical for each cell) and ensure that most spots identified as NPCs are indeed double-labelled.

2d) What would be the impact of mild alterations in NPC distribution, leading to more undistinguishable clusters ? are these taken in account in the quantification and modeling?

3) While conclusions regarding NPC density, NPC clustering and NPC basket alteration in the vicinity of the nucleolus mainly comfort previous studies in budding yeast, no mechanistic insights are provided.

3a) The authors propose that NPC density is maintained throughout cell cycle... "by a mechanism that restricts the assembly of new NPCs". However, the latter statement is solely based on the normal NPC density observed in two autophagy mutants. Can the authors exclude that another disassembly mechanism could be involved ? (for instance, at a given stage of mitosis or in the midzone, as they propose later ??). In addition, the authors do not address the control mechanism and solely discuss the unlikely involvement in fission yeast of the TPR/Erk-dependent pathway reported in mammalian cells.

3b) In Fig 1E, it appears that a significant (*) decrease in NPC density occurs between early and late mitosis. How do the authors interpret this ? Related to this aspect, what is the p-value between late mitosis and early G2? (and between G1/S versus early G2) ?

Minor points:

- Line 65-69: This sentence might be rephrased since the discrepancy between the data from [Kittisopikul et al., 2021 and Boumendil et al., 2019] on one hand, and from McCloskey et al., 2018 on the other, does not concern the ERK pathway (only studied by McCloskey et al), but rather the impact of TPR depletion on NPC assembly (found to cause an increase in NPC density/assembly in McCloskey et al. but not in the two other studies).

A more appropriate way to present might be: "However, the ubiquitous role of TPR for negative regulation of NPC density has been debated (Kittisopikul et al., 2021; Boumendil et al., 2019)".

- Line 85: "suggesting that either the SPB itself or associated factors control NPC recruitment OR ASSEMBLY"

If they wish to keep the word "ASSEMBLY", the authors should clarify that the possible implication of SPB in NPC assembly is their own hypothesis as Ruthnick et al rather suggested a "a mechanism of active recruitment of NPCs to the inserting SPB."

- Fig 1A. Graph: it would be clearer if an arrow were drawn to represent the FWHM

- Fig 1B: some Nups mentioned in this study (Nup44, Nup124) are not listed in this figure.

- line 138 and Fig 1E: "the number of NPCs also increases through interphase to maintain a constant NPC density". The fact that these data are solely based on Nsp1-GFP localization should be clearly stated.

- line 149 : "In agreement with these findings, we observed NPCs undergoing disassembly in the anaphase bridge midzone region". The term "disassembly" does not seem appropriate here. Indeed, this statement is solely based on the lack of Nup60 but not Nup37 or Cut11 in the midbody. Yet, this phenotype may also reflect the existence of two subsets of NPCs in the pre-existing nuclei, or a specific behaviour in the midzone of Nup60 (whose *S. cerevisiae* ortholog is a dynamic Nup), not reflecting NPC disassembly.

To address this issue, the authors should determine if all Cut11-labelled NPCs are positive for Nup60 on the NE of G2/M nuclei. They should also analyze other Nups localization at midzone in late M to determine the extent of this possible disassembly.

- line 298: "Exclusion of NPCs near the SPBs requires Lem2-mediated centromere tethering". As the authors did not assess the impact of Sad1 mutation on NPC distribution, and since the main phenotype is observed in the *csi1Δ* mutant, the title should be modified towards "Exclusion of NPCs near the SPBs requires Lem2 and centromere tethering".

Reviewer #2 Review

Comments to the Authors (Required):

Varberg et al, perform a detailed quantitative analysis of NPC number, distribution and composition in the fission yeast nucleus by using EM, 3D-SIM, and confocal live microscopy. They also follow different approaches to quantify and determine NPC behaviour in different conditions. They show that NPC density is maintained in cells with different nuclear size and is also constant during the cell cycle. NPC formation is dependent and/or coordinated with membrane expansion. They identify different nuclear envelope domains based of their reduced NPC density and different NPC composition. They find reduced NPC density around the SPB and around the nucleolar domain which also shows NPCs lacking the nuclear basket. In the SPB, the exclusion

of NPCs depends on centromeric chromatin attachment by Lem2 and other factors, demonstrating that the attachment of chromatin influences NPC distribution.

I think this study has been nicely executed and sets the basis for future studies aimed at understanding important questions regarding NPC biology. In addition, no such characterization has been done in the fission yeast before. Fission yeast has emerged as a valuable model to study NPC function, organization and dynamics and recent studies including this one, are uncovering a high level of heterogeneity and structural plasticity of NPCs which open new and exciting questions for future studies. For all these reasons I think this study fits this journal and I recommend publication.

There are however, a number of issues and concerns listed below that should be addressed to improve or clarify some points along the manuscript.

Major points:

1) Pag9 line 173. The authors find that there is variability in the number of NPCs when using different tagged Nups. This is shown in Figure 1E and S1E. However, it is difficult to see it clearly due to overlapping of colored dots. I think this piece of information should be discussed in the manuscript. What is the extent of the difference? Which tagged nups show the bigger or smaller number of NPCs?.

This difference might be due to a lack of functionality of certain tagged nups affecting NPC assembly. Does this difference correlate somehow with the frequency of NPC clustering?. Alternatively, this difference might reflect a true different functional composition of NPCs or different intermediate states of NPC assembly. Is there any correlation between the position of any given nup in the NPC structure with the number of NPC detected with this tagged nup?.

Throughout the manuscript the authors use mainly Nsp1 and occasionally Nup37 and Nup97 as nup marker. The authors should at least, briefly justify why they choose these, among all the nups tested.

2) In page 8 line 164: the authors present data on average number of NPCs, nuclear surface and NPC density in mid-G2 cells. It is not clear which tagged nups have been used to obtain these data. Are these measurement the average of all tagged-nups shown in Figure 1 B?, are these values from Nsp1 (FigS1D)? or are these the average of measurements obtained using Nsp1 and Nup97 (FigS1E)?. The authors should clearly indicate which nups have been averaged to obtain these parameters. Figure S1E legend should indicate the meaning of the red and blue dot.

3) In this work, the authors clearly demonstrate that chromatin tethering by Lem2 is responsible for the exclusion of NPCs around the SPB region. It has been recently shown that Lem2 also binds to non-transcribed intergenic spacers of rDNA repeats and *lem2Δ* cells show fragmented nucleoli, suggesting that Lem2 anchors rDNA to the NE (Iglesias et al, Mol Cell 2020. PMID: 31784357). It would be interesting to test whether Lem2 also influences NPCs distribution at the nucleolar region. Have the authors analysed Lem2 distribution at the NE over the nucleolus?. With the data already in their hands, the authors could analyse NPC distribution at the nucleolar region in *lem2Δ* cells.

In the same study from Moazed's lab (Iglesias et al, Mol Cell,2020), they show that whereas Lem2 binds the central core of centromeres, several nups including the basket nup, Nup211 bind heterochromatin at pericentromeric repeats. Thus, the SPB exclusion zone might reflect this differential pattern of chromatin binding of Lem2 and NPCs. I think the work from Iglesias et al, should be cited in the manuscript.

Minor points

1) In different parts of the manuscript the authors refer to different aspects of SPB biology in *S. cerevisiae* and *S.pombe*. For example, in line 249-251, or line 431 they discuss SPB insertion in *S.cerevisiae* vs *S.pombe* and in Figure 5A they show the distance from the SPB to the nearest NPC in single, duplicated or inserted SPBs.

I think is important to point out that the SPB in *S.cerevisiae* is inserted into the NE whereas in *S.pombe* is cytoplasmic during interphase and inserted upon mitotic entry. This is mentioned somehow in the discussion section, however I think it would be helpful to include this information earlier.

2) Pag 7 line 139. The authors observe occasional differences in NPC density between sister nuclei. They go on and determine that this asymmetry is random and does not correlate with the inheritance of the mother or daughter SPB. The authors should show the % of cell divisions in which this asymmetry is observed. I am wondering whether the authors have analysed the outcome of this phenotype, do these cells recover the normal NPC density? Do they maintain the average NE surface volume? Do they have altered rate of growth?

3) In Figure 1E, the authors show quantitation of NPC number, surface area and NPC density using different nups. The meaning of the red, purple, blue and green dots should be explained in figure legend.

4) The clustering of NPCs is a striking and very interesting phenotype and the fact that it appears occasionally in wild type cells suggest that it is functionally relevant. These clusters might represent sites of NPC assembly, sites of NPC disassembly or just random aggregates of defective NPCs. Have the authors analysed NPC clustering by using different tagged nups? If so, have they detected differences in NPC nup composition in these clusters/rings that might give a clue of the state of these NPCs? I find that the discussion of this phenotype could be extended. For example, nup132 orthologues have been shown to be required for proper NPC assembly, which favours the idea that the clusters observed in nup132 mutants might be defective NPCs. Nem1 however is known to affect lipid synthesis and might affect NPC distribution but in principle not NPC structure. The authors find that in the case of *nem1Δ* daughter nuclei had normal morphology and NPC density after mitosis, however they don

ˆt explain what happens in the case of nup132Δ nuclei after mitosis. What happens in the next cell cycle with these big NPC clusters?

5) From the text I understand that the exclusion zone is studied in duplicated non inserted SPBs (Figure 5). However this exclusion is observed throughout the cell cycle. There is a point at the beginning of mitosis where the nucleoporin Cut11 localizes to the SPBs. In microscopic images Cut11 seems to "cover" the SPBs coincident with SPB insertion into the NE. I am wondering whether the authors have analysed the exclusion zone using tagged-Cut11. Does Cut11 show the exclusion zone around the SPB? Is the exclusion maintained during the period of SPB insertion? (when Cut11 accumulates at the SPBs).

6) Pag 19 line 390: "Spk1, the fission yeast ERK orthologue, is not essential for cell viability (Toda et al, 1991), suggesting that mechanistic details of this pathway may differ between species". I think that this statement is misleading and should be rephrased, as Sty1 and Pmk1 are the two main MAPKs operating during vegetative growth in *S.pombe*.

7) Page 11. Fig3C, nup132Δ cells grown on solid media show increased NPC ring number. Based on that, in page 19, line 399, the authors suggests that changes in nutrient availability triggers NPC reorganization into ring clusters of a consistent size. However it is not clear whether aged wild type cells also show increased NPC ring number. Does this happen also in aged wt and nem1Δ cells? Also the fact that spores that are kept in low nutrient media and are mostly metabolically inactive show normal NPC density and distribution argues against the former suggestion.

8) In page 21 line 445: " However, the fact that Nup exclusion is decreased at the SPB when NPCs are artificially tethered suggests that an SPB-derived signal such as phosphorylation by a SPB-localized kinase likely does not induce NPC remodeling in this region of the NE".

I do not agree with this. The authors are forcing the localization of NPCs/Nsp1 to the SPB and this attachment is not expected to be broken by any signal or phosphorylation. The authors could explore the composition of these artificially bound NPCs to test whether they are 'complete NPCs' or whether they lack specific nups that might suggest that they are undergoing any type of modification/disassembly.

Reviewer #3 Review

Comments to the Authors (Required):

Varberg and colleagues report a very detailed quantitative analysis of nuclear pore complex (NPC) distribution in *Schizosaccharomyces pombe*. Applying SIM to fixed yeast cells expressing endogenously GFP-tagged nucleoporins, the authors achieve near single NPC resolution and find that NPC density is maintained constant across the cell cycle. NPC density is also roughly constant despite manipulation of cell size by various genetics means. Induction of NPC clustering by mutation of either nup132 or nem1 reveals the existence of ring-like structures with a diameter of ~250-280 nm, i.e. about 2-2.5 times the diameter of NPCs. The nature of these structures is not further described. Next, the authors show that NPC density is reduced in proximity of the nucleolus, similarly to what has been described in other organisms. The authors also investigate NPC density in vicinity of the spindle pole body (SPB) and find that, in contrast to the situation in budding yeast, NPCs are excluded in a zone of ~200 nm surrounding the SPBs. Interestingly, this exclusion zone is reduced to approximately half the size in lem2 mutants and absent in csi1 mutants that lack centromere tethering. To address the functional relevance of NPC exclusion from the immediate vicinity of the SPBs, the authors employ an inducible tethering system that brings NPCs (or at least nucleoporin Nsp1) closer to the SPB. This causes a mild growth defect and alterations in mitotic spindle orientation and mitotic progression.

The study is very convincing based on the quality of the data and the large number of observations in most experiments. The M&M section is rich on details and the manuscript is very well written. The conclusions are mainly descriptive and in remains mechanistically unclear why NPCs need to be kept away from SPBs, in particular considering that the situation seems to be reversed in budding yeast.

I only have minor specific points:

Explain FWHM; is used first time in line 129 and defined in line 566.

The preferential localisation of NPC clusters to SPBs mentioned in lines 229-230 is not clear from Figure 3F. Please provide also merged images of the two channels.

Ppc89 is used as marker in many images: explain that Ppc89 is a SPB component.

Nsp1 is indicated as component of the NPC central channel in Figures 1B and 5F but then as NPC basket protein in Figure 7A. This is a bit confusing. The dual localisation is explained in lines 345-346. I suggest to give this explanation earlier.

The tethering experiment in Figure 7 would be strengthened considerably if the authors could demonstrate that other nucleoporins (and thus presumably the entire NPC) also change localisation and not just the tethered protein itself (Nsp1).

February 24, 2022

Re: Life Science Alliance manuscript #LSA-2022-01423-T

Dr. Joseph M Varberg
Stowers Institute for Medical Research
1000 E. 50th St
Kansas City 64110

Dear Dr. Varberg,

Thank you for submitting your manuscript entitled "Quantitative analysis of nuclear pore complex organization in *Schizosaccharomyces pombe*" to Life Science Alliance. I encourage you to submit a revised manuscript addressing Reviewer 1's comments, without the need for additional experimentation.

The typical timeframe for revisions is three months.

Thank you for this interesting contribution to Life Science Alliance. We are looking forward to receiving your revised manuscript.

Sincerely,

B. MANUSCRIPT ORGANIZATION AND FORMATTING:

We appreciate the attention given by the reviewers to the revised manuscript and were glad to see that Reviewer 2 felt that we addressed all of their questions and concerns. We have made additional modifications to the manuscript text to address the remaining comments from Reviewer 1, which are outlined below.

Reviewer #1 (Comments to the Authors (Required)):

In this revised version, the authors have followed the suggestions of the reviewers to clarify their previous manuscript, and took in account my minor comments. In contrast, the additional experiments performed do not address my major points in a satisfying manner. Hence, the resulting publication now mainly provides a properly-performed description of NPC distribution in *S. pombe*.

We are glad that the reviewer agrees that our manuscript is a properly performed description of NPC distribution in *S. pombe*.

1) I appreciate that the authors made the effort to perform the critical control required for the tethering experiment (answer to my major point 1). Unfortunately, this control experiment demonstrated that their tethering approach was not appropriate; hence the authors have removed the Figure 7 of the previous ms. They indicate that they "hope to revisit this approach in future work after optimization of the tethering system". Therefore, the revised manuscript becomes even more descriptive than the previous version.

Indeed, there is no more data demonstrating, as still stated by mistake in the abstract of the revised version, that NPC exclusion around the SPB... is important for timely mitotic progression.

Thank you for pointing out this oversight in the abstract. It has now been corrected.

While removing this set of potential functional data, the authors do also not perform one simple experiment suggested by reviewer 2 (in his main point 3), that would have been to analyse whether Lem2 influences NPC distribution at the nucleolar region as it does underneath the SPB. Although the authors had all tools in hand to address this question, this was considered by the authors to extend beyond the context of their current manuscript.

Yet, this would perhaps have help to determine the extent to which centromeres as such, rather than general chromatin crowding, influence local NPC distribution. In that case, the conclusion of their abstract that "interaction between the centromere and the NE" influences local NPC distribution should be (or not) modified towards "interaction between chromatin and the NE" influences local NPC distribution".

We appreciate the suggestion but for the reasons outlined below we decline to add these additional experiments at this time. As we mentioned in our previous response, we are actively conducting experiments to determine how INM proteins, including Lem2, may influence the density of NPCs over the nucleolus. However, looking simply at *lem2Δ* for the effects on NPC distribution over the nucleolus does not, on its own, allow us to properly address the question with respect to the influence of chromatin as suggested by the reviewer. Lem2 has multiple reported functions beyond interaction with heterochromatin, including in regulating membrane flow into the nuclear envelope to control nuclear size (Kume et al, Nature Communications 2019; Hirano et al, Communications Biology 2020), and regulating nuclear envelope integrity (Gonzalez et al 2012). Additionally, Lem2 is just one of many proteins known to interact with and influence the localization of heterochromatic regions to the NE

periphery: others include Ima1 and Man1 (Steglich et al, Nucleus 2012; Gonzalez et al, Nucleus 2012), Amo1 (Holla et al, Cell 2020; Charlton et al, PNAS 2020), Taz1 (Cooper et al, Nature 1998), Bqt3/4 (Ebrahimi et al, eLife 2018; Chikashige et al, JCB 2009) and Fft3 (Steglich et al, PLoS Genetics 2015). For these reasons, any phenotype observed for *lem2Δ* could not be attributed to effects on changes in heterochromatin organization at the periphery without additional experiments ruling out effects on the nuclear envelope. Similarly, addressing the potential role for tethering of telomeric and/or subtelomeric chromatin on NPC density over the nucleolus would require experiments beyond simply looking at *lem2Δ*. For these reasons, we feel that this is beyond the scope of the current manuscript.

2) Regarding the NPC quantification methodology (main points 2 and 3)

In the revised version, the authors now better clarify the limit of their quantitative "NPC number" analysis.

- To estimate the extend of NPC undercounting, the authors had simulated randomly distributed NPCs, and show that in the range of NPC density they analyze in *S. pombe*, [but also of nuclear size as included in this revised version] the underestimation "fell within 10-30% of the true simulated value".

However, it becomes even clearer in the revised version (see the statistics of the last two panels of Figure 1E) that NPC undercounting can also be caused by the non-random distribution/clustering of NPCs that could not be taken in account in their model.

As such, the authors apparently rather trust in figure 1E the conclusions of their alternative quantitative approach, namely density/ unit surface measurements that indicated no significant difference between the NPC G1/S cells and early to late G2 cells; while the difference is very significant ($p < 0.01$, **, or perhaps $< 0.005 = ***$) when NPC "numbers"/surface is measured. To my point, this partly dismisses the interest of this sophisticated "individual" NPC approach the manuscript initially focused on.

We agree that SIM and other microscopy methodologies with similar resolution will undercount NPCs and have stated this on **line 162 of the manuscript** ("Despite the improved lateral resolution offered by SIM..."). Short of using light-microscopy methods that can resolve all individual NPCs including those in clusters without error, there will inevitably be undercounting of NPCs. We have applied our analysis methods to simulated data to provide insight into the estimated magnitude of undercounting, which is **shown in Figure S1G and discussed in lines 162-172**. Additionally, we have used an orthogonal method that does not rely on resolution of independent pores (**shown in Figure 1E and Figure S2A-D**), which we believe is an appropriate method to limit misinterpretation of our SIM results. Despite the resolution limits for SIM, we believe it is still a useful tool to provide quantitative information about NPC number and to identify features of NPC distribution in relatively high throughput. This enables experiments that would otherwise be unfeasible using other higher-resolution imaging approaches such as serial thin-section EM. As instrumentation and analysis tools continue to improve for SIM and related light-microscopy techniques, approaches like we have used in this manuscript will allow for more sophisticated and sensitive analysis of mechanisms influencing NPC number and distribution.

2b) By performing additional experiments, the authors confirm and better illustrate "the variations in NPC number detected using different tagged Nups" (raised in my main point 2b). However, they do not address the question of whether these differences could be caused by a mild clustering phenotype (apparent changes in density) or by a possible heterogeneity in NPC composition (as for instance observed in an extreme manner for the basket protein Alm1 _ new panels 4E-G). Indeed, the authors choose to perform dual color analysis using a novel set of Nups (Nup40-mCh and Nup44-GFP). Dual

imaging of Nup97 or Nup60 (average density 4 NPC/ μm^2) versus Nsp1 (average density 6 NPC/ μm^2) would have been more pertinent.

The reviewer previously recommended using two-color SIM as a method to confirm that the points that we were identifying using our analysis approach were bona fide NPCs. We chose to examine two Nups that were members of the same complex and are added at similar times during NPC assembly (based on the kinetics of their orthologs during budding yeast NPC assembly in Onischenko et al, Cell 2020), for which we would expect to see a high degree of co-localization (Nup40 and Nup44). This confirmed that our method was correctly identifying true pores – that is, spots that were positive for both Nups. The Nsp1/Alm1 SIM data confirmed that our methods can derive a quantitative measure of heterogeneous NPC composition. Interpretation of the Nsp1/Alm1 SIM data is made possible by our complimentary experiments showing heterogeneity over the nucleus (i.e., we knew we should see a population of pores lacking Alm1), as well as work on the orthologous Mlp1/Mlp2 basket Nups in budding yeast. It is almost certain that some of the variability in number seen for different Nups is attributable to heterogeneity in composition, including at the very least, heterogeneity arising from pores that are actively being assembled. The data from Onischenko et al. showed that NPC assembly takes over 45 minutes in *S. cerevisiae* from initiation to full NPC maturation, with certain Nups arriving at specific stages. Additional pairwise two-color SIM imaging will likely be a useful approach to address the extent and potential source of heterogeneity we observe in NPC numbers for different Nups.

Minor point:

I do not think that normalizing the NPC density plot in Figure 2 within each individual experiment/panel "avoids any potential confusion". Rather, it partially masks the variability caused by Nups tagging. Note also that if normalized, the value plotted should be in arbitrary units and not NPC number/ μm^2 .

Thank you for pointing out the error in axis labeling, which has been fixed. We appreciate the reviewers' comment. However, as the earlier presentation without normalization caused confusion, we feel that the normalization helps ensure that the readers' focus is on the comparisons between groups for each panel. We are not attempting to "mask variability caused by Nups tagging" and have been upfront about highlighting these differences in the text **at lines 181-188** ("While similar trends in NPC number and density were observed using multiple tagged nucleoporins (**Figure S1E**), variability in the number of NPCs detected was observed....) and in **panels E and H in Supplemental Figure 1**.

March 4, 2022

RE: Life Science Alliance Manuscript #LSA-2022-01423-TR

Dr. Joseph M Varberg
Stowers Institute for Medical Research
1000 E. 50th St
Kansas City 64110

Dear Dr. Varberg,

Thank you for submitting your revised manuscript entitled "Quantitative analysis of nuclear pore complex organization in *Schizosaccharomyces pombe*". We would be happy to publish your paper in Life Science Alliance pending final revisions necessary to meet our formatting guidelines.

- please add ORCID ID for secondary corresponding author-they should have received instructions on how to do so
- please add a conflict of interest statement to your main manuscript text
- please consult our manuscript preparation guidelines <https://www.life-science-alliance.org/manuscript-prep> and make sure your manuscript sections are in the correct order
- please add your main, supplementary figure, and table legends to the main manuscript text after the references section
- please use the [10 author names, et al.] format in your references (i.e. limit the author names to the first 10)
- please add a callout for Figure 1D to your main manuscript text

FIGURE CHECKS:

- some of the panels in Figures 5 and 6 appear blurry. Please make sure these images are as clear as they can be.

A. FINAL FILES:

B. MANUSCRIPT ORGANIZATION AND FORMATTING:

Sincerely,

March 14, 2022

RE: Life Science Alliance Manuscript #LSA-2022-01423-TRR

Dr. Joseph M Varberg
Stowers Institute for Medical Research
1000 E. 50th St
Kansas City 64110

Dear Dr. Varberg,

Thank you for submitting your Research Article entitled "Quantitative analysis of nuclear pore complex organization in *Schizosaccharomyces pombe*". It is a pleasure to let you know that your manuscript is now accepted for publication in Life Science Alliance. Congratulations on this interesting work.

DISTRIBUTION OF MATERIALS:

Again, congratulations on a very nice paper. I hope you found the review process to be constructive and are pleased with how the manuscript was handled editorially. We look forward to future exciting submissions from your lab.

Sincerely,
